# Mechanism of integrin activation by talin and its cooperation with kindlin

Fan Lu[1,2,4], Liang Zhu[1,4], Thomas Bromberger[3,4], Jun Yang[1], Qiannan Yang [1], Jianmin Liu [1], Edward F. Plow[1], Markus Moser [3✉] & Jun Qin [1,2✉]

Talin-induced integrin binding to extracellular matrix ligands (integrin activation) is the key step to trigger many fundamental cellular processes including cell adhesion, cell migration, and spreading. Talin is widely known to use its N-terminal head domain (talin-H) to bind and activate integrin, but how talin-H operates in the context of full-length talin and its surrounding remains unknown. Here we show that while being capable of inducing integrin activation, talin-H alone exhibits unexpectedly low potency versus a constitutively activated full-length talin. We find that the large C-terminal rod domain of talin (talin-R), which otherwise masks the integrin binding site on talin-H in inactive talin, dramatically enhances the talin-H potency by dimerizing activated talin and bridging it to the integrin co-activator kindlin-2 via the adaptor protein paxillin. These data provide crucial insight into the mechanism of talin and its cooperation with kindlin to promote potent integrin activation, cell adhesion, and signaling.

[1] Department of Cardiovascular & Metabolic Sciences, Lerner Research Institute, Cleveland Clinic, 9500 Euclid Ave., Cleveland, OH 44195, USA. [2] Department of Biochemistry, Case Western Reserve University, Cleveland, OH 44106, USA. [3] Institute of Experimental Hematology, School of Medicine, Technische Universität München, Munich D-81675, Germany. [4]These authors contributed equally: Fan Lu, Liang Zhu, Thomas Bromberger. ✉email: m. moser@tum.de; qinj@ccf.org

The attachment of cells to the extracellular matrix (ECM) is a fundamental cellular process crucial for the development and responses of multicellular organisms, and its dysfunction has been linked to many human disorders such as cancer, stroke, and thrombosis[1–3]. This cell-attachment process, also called cell–ECM adhesion, is critically controlled by integrins, a class of heterodimeric (α/β) cell-surface receptors discovered three decades ago[4–6]. Each integrin subunit consists of a large extracellular domain (ectodomain), a single transmembrane domain (TM), and typically, a small cytoplasmic tail (CT). Integrins can adopt different conformations that display different affinities for ligand. The transition from low- to high-affinity ligand-binding state, referred as integrin activation, can be triggered by a distinct process called integrin inside-out signaling where cellular stimulation elicits intracellular signal(s) to perturb the integrin cytoplasmic face, inducing a global conformational change of the receptor and its subsequent high-affinity binding to ECM ligand[7]. Ligand binding to integrins also transmits signals in an "outside-in" manner, leading to the assembly of large integrin-cluster-containing complexes involving hundreds of proteins (focal adhesions, FAs) that are linked to actin filaments. These adhesion complexes form signaling hubs that can transmit signals to promote assembly–reassembly–disassembly cycles of FAs, leading to cytoskeleton rearrangement and dynamic adhesion-dependent processes such as cell migration and spreading[7].

As the first key step to trigger cell–ECM adhesion, integrin activation has been extensively studied for more than two decades, which led to the discovery of talin as a major integrin activator[7–13]. Talin is a large cytoskeletal protein containing an N-terminal FERM-like head domain (talin-H, 1–405), a linker (406–485), and a C-terminal rod domain (talin-R, 486–2541)[14] (Fig. 1a). Talin-H, which constitutes less than 20% of the molecule, contains four subdomains, F0, F1, F2, and F3, whereas the large talin-R can be subdivided into 13 helical bundle subdomains (R1–R13) followed by a C-terminal dimerization domain (DD) (Fig. 1a). In unstimulated cells, talin is randomly distributed in the cytosol in an autoinhibited conformation where the integrin-binding site on talin-H is masked by a talin-R subdomain (talin-R9, see Fig. 1b)[15–18]. Upon stimulation, talin is recruited to the membrane[19–22] and activated by phosphatidylinositol 4,5-bisphosphate (PIP2)[15,17,18], which is locally enriched by PIP2-producing kinase PIPKIγ that is recruited by talin[23,24]. Talin activation allows talin-H binding to the integrin-β CT[15–18], which in turn disrupts the integrin-α/β CT association to trigger the conformational activation of the receptor[25,26]. While this talin-H-induced integrin-activation scheme has been widely adopted in the literatures (Fig. 1b)[7–13], how talin-H actually acts in the context of full-length talin remains undefined. Curiously, talin-H alone is much less potent in inducing integrin activation than specific mutations or truncation of the integrin CTs or treatment with the divalent cation $Mn^{2+}$ [27–29], suggesting that talin-H requires additional factors to potentiate integrin activation. Moreover, ECM ligands are typically multivalent/multimeric[30] such as fibronectin (dimeric)[31], fibrinogen (dimeric)[32], collagen (trimeric)[33], and Sars-Cov-2 Spike protein (trimeric)[34], and thus a complete integrin-activation process should involve the conformational change of multiple integrins that are organized into a microcluster[35] to bind a multivalent ligand with high affinity[30], but the molecular mechanism underlying integrin microclustering remains unknown. Extensive search for regulators of integrin activation led to the discovery of kindlins, a subclass of FERM-domain proteins (FERMTs) consisting of kindlin-1, -2, and -3 [9–13,36,37]. Kindlins have also been shown to bind to integrin-β CT and enhance talin-H-mediated integrin activation[9–13,36,37]. However, unlike talin-H, kindlin does not disrupt integrin-α/β CT association and induce integrin

activation itself[38]. Rather, kindlin may somehow enhance talin-H-mediated integrin binding to multivalent but not monomeric ligands[39]. The mechanism of how kindlin cooperates with talin-H to promote integrin activation remains highly elusive, despite extensive investigation over the past 15 years[38–44].

In this study, we set out to investigate these major puzzling issues through combined biochemical, structural, and cell biological approaches. We found that while being capable of activating integrin, talin-H alone exhibits unexpected low potency to induce integrin binding to multivalent ligand compared with a constitutively activated full-length talin. This finding challenges the long-standing talin-H "only" model for integrin activation (Fig. 1b) and prompted us to perform a comprehensive investigation, which unveiled a crucial role of talin-R in regulating talin-H-mediated integrin activation. Specifically, we found that talin-R, which otherwise masks the integrin-binding site on talin-H in inactive talin, dramatically enhances the talin-H-mediated integrin activation by dimerizing active talin and bridging it with the integrin co-activator kindlin-2 via the adaptor protein paxillin. Mutations disrupting this talin-R-mediated complex network that diminished integrin activation, ultimately led to substantially reduced cell adhesion and spreading. Our study significantly advances the understanding of the terminal step of integrin activation, highlighting a crucial role of talin-R in forming a crucial talin dimer/paxillin/kindlin complex to cluster integrins, thereby promoting potent integrin binding to multivalent ligands, focal adhesion formation, and cell adhesion.

## Results

**A constitutively activated full-length talin protein promotes strong integrin activation.** As described above, release of talin autoinhibition enables talin-H to bind and activate integrin (Fig. 1b). In this model, talin-R was assumed not to be involved in integrin activation, meaning that talin-H should have a similar capacity to activate integrins compared with an activated (non-autoinhibited) full-length talin. To vigorously evaluate this model, we designed an activated full-length talin mutant. Instead of single-point mutations that may only partially disrupt talin autoinhibition[15,16], we simultaneously mutated three critical autoinhibitory interface residues M319, T1767, and E1770 (M319A, T1767L, and E1770K, referred to active talin or tlnM3, Fig. 1b, c) based on the crystal structure of the autoinhibitory talin-F2F3/talin-R9 complex (PDB code 4F7G)[17]. Supplementary Fig. 1a shows GST pull-down experiments, confirming our design in that tlnM3 but not talin-WT, binds to integrin-β3 CT. Consistently, tlnM3 eluted earlier than talin-WT in size-exclusion experiment (Supplementary Fig. 1b and 1c) likely due to its extended shape compared with the autoinhibited/inactive talin as also suggested by the recent cryo-EM analysis[18]. Surface plasmon resonance (SPR) experiments indicated an affinity of integrin-β3 CT binding to tlnM3 at $K_D \sim 14.5$ nM (Supplementary Fig. 1d). Under the same experimental conditions, this affinity is slightly higher than that of integrin-β3 CT to talin-H ($K_D \sim 29.7$ nM, Supplementary Fig. 1e) possibly because full-length talin may form dimer[45] that strengthens integrin binding (of note, a talin-deletion mutant without C-terminal dimerization domain binds integrin similarly as talin-H with $K_D \sim 36.0$ nM, see Supplementary Fig. 1d). Next, we compared the effect of talin-H and tlnM3 on the ligand binding to prototypic integrin αIIbβ3 stably expressed in Chinese-hamster ovary (CHO) cells—a well-established assay for studying integrin activation[46]. To our surprise, while tlnM3 only slightly increased binding of integrin αIIbβ3 to the monovalent ligand fibronectin repeat 10 (FN10) compared with talin-H alone (Fig. 1d), tlnM3 dramatically increased integrin-αIIbβ3 binding to PAC-1—a well-known

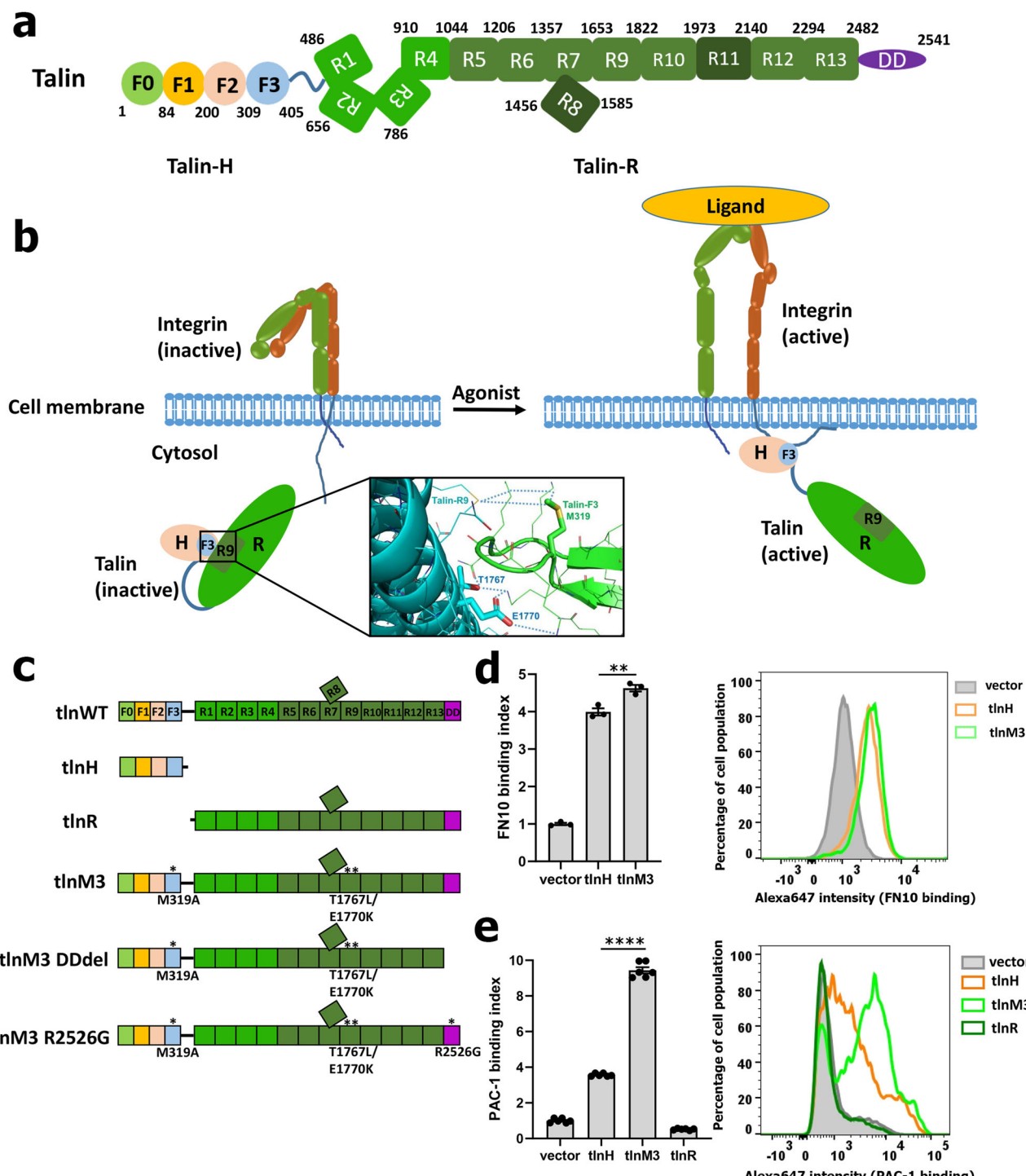

**Fig. 1 Talin is an intracellular activator of integrin. a** Domain organization of talin. N-terminal head domain (talin-H) is composed of four subdomains and C-terminal rod domain (talin-R) contains 13 subdomains followed by dimerization domain (DD). Talin-H and talin-R are connected by a flexible linker. **b** A scheme of talin activation and talin-mediated integrin activation. Upon stimulation, inactive talin is released from an autoinhibited state to expose talin-H to bind integrin cytoplasmic tail and trigger conformational change of integrin to bind ligand. The autoinhibitory interface between talin F3 and talin R9 is displayed in the box showing how M319, T1767, and E1770 are involved in the interface and mutated to release the autoinhibiton. **c** Diagram of the various talin variants, including mutation sites to activate talin or disrupt talin dimerization. **d** Left panel: Integrin-activation level reflected by monomeric FN10 binding is not elevated much by constitutively activated full-length talin mutant, (tlnM3) compared with talin-H (tlnH). **p = 0.0081 with 95% confidence interval 0.2727–0.9914 (t-test), N = 3 biologically independent samples. All values are given as mean ± S.E.M. Right panel: FN10 staining was similar between tlnH (orange) and tlnM3 (light green). **e** Left panel: Integrin-activation level reflected by PAC-1 binding is substantially elevated by tlnM3 compared with tlnH. Talin rod (tlnR) alone has no effect on integrin activation. ****p < 0.0001 with 95% confidence interval 5.478–6.260 (t-test), N = 6 biologically independent samples. All values are given as mean ± S.E.M. Right panel: Significantly more PAC-1 staining was induced by intact active talin (tlnM3, light green) in comparison with talin head (tlnH, orange) without normalization. **d**, **e** Raw data are provided in Source Data file.

ligand-mimetic multivalent antibody that recognizes activated αIIbβ3[47] compared with either talin-H alone (Fig. 1e) or full-length WT talin (Supplementary Fig. 1f). Because expression of talin-R alone had no effect on PAC-1 binding (Fig. 1e), the drastic effect of tlnM3 compared with talin-H on PAC-1 binding (Fig. 1e) suggests a previously unrecognized but crucial role of talin-R in regulating integrin binding to multivalent but not monomeric ligands. As mentioned earlier, multivalency is an important feature of most of physiological integrin ligands.

**Talin-R-mediated talin dimerization promotes integrin binding to multivalent ligands.** Talin exists in a monomer–dimer equilibrium at native conditions[45,48], and prefers dimerization at higher protein concentration (>3 μM)[45] which may be reached in cells with high talin levels up to 50 μM[49] and can be further elevated by subcellular enrichment after recruitment to the plasma membrane. A compact-shaped monomeric structure of autoinhibited talin-WT has been recently solved at lower salt concentration (75 mM) and low protein concentration (~1 μM) favoring the monomeric state[18], although a significant dimeric population was also observed under these conditions (see Fig. 2c in Dedden et al. 2019[45]). SPR experiments by immobilizing tlnM3 onto the CM5 chip and then flowing through tlnM3 in 150 mM salt revealed the distinct response curves that validated the intermolecular tlnM3/tlnM3 interaction with $K_D$ ~ 1.82 μM (Supplementary Fig. 2a). This affinity is consistent with the earlier sedimentation studies on talin dimerization (at ~ 3 μM)[45], which used higher salt (220 mM NaCl) to mimic the talin-activation status (elongated state)[18]. Previous studies revealed that talin dimerization is critically mediated by the C-terminal talin-R dimerization domain (DD) (Fig. 1a)[50] and its deletion was shown to disrupt the dimerization of full-length talin-WT (see Fig. 2C in Dedden et al. 2019[45]) and affect focal adhesion formation and cell spreading[51]. Since dimeric talin could bind and bridge two integrins, thereby promoting the formation of integrin micro-clusters, we wondered if the talin dimerization could account for the enhanced multivalent PAC-1 binding induced especially by full-length activated talin (see the scheme in Fig. 2a). To examine this possibility, we generated a tlnM3 construct lacking the C-terminal DD domain (tlnM3 DDdel as illustrated in Fig. 1c). Remarkably, Fig. 2b, c show that deletion of the talin-R DD domain significantly impaired tlnM3-induced PAC-1 binding to integrin-αIIbβ3. Moreover, a specific point mutation disrupting talin dimerization (R2526G[50,51]) also significantly impaired tlnM3-induced PAC-1 binding (Supplementary Fig. 2b). To further evaluate the role of talin dimerization in regulating the integrin/ligand interaction, we took advantage of our established talin1 and talin2 double-deficient fibroblast system (talin1/2dko)[19,21]. These cells were retrovirally transduced with constructs encoding for C-terminally ypet-tagged full-length talin and talin constructs that either lacked the DD domain or contained the dimerization-disrupting R2526G mutation. Remarkably, integrin binding to the multivalent ligands, fibronectin, vitronectin, and laminin were all reduced by the mutations that blocked talin dimerization (Fig. 2d, Supplementary Figs. 2c, d), indicating for the first time that talin dimerization plays a critical role in controlling integrin binding to multivalent ligands. However, the talin DD deletion mutant and the R2526G mutant only impaired ligand binding partially (~25%) compared with intact tlnM3, and were still significantly more potent than talin-H (Fig. 2b, c, Supplementary Fig. 2b), suggesting the existence of additional factors that control talin-R-mediated integrin binding to multivalent ligands.

**Talin-R bridges talin with kindlin-2 via paxillin to promote integrin binding to multivalent ligands.** Next, we investigated

whether talin utilizes talin-R to promote integrin activation by cooperating with other integrin regulators. Given the essential role of kindlins as integrin co-activators, and the fact that kindlins also enhance talin-H-mediated PAC-1 binding but not FN10 monomer binding[39], which is similar to the tlnM3 function in Fig. 1d, e, we wondered whether talin-R plays a role in linking talin with kindlin-2—the most widely expressed member of the kindlin family[9–13,36,37]. Interestingly, kindlin-2 drastically enhanced the capacity of tlnM3 (Fig. 3a, b) but not tlnWT (Fig. 3c) to induce PAC-1 binding of integrin-αIIbβ3. Of note, this enhancement was even greater than the known effect of kindlin-2 on talin-H-mediated PAC-1 binding to integrin-αIIbβ3[28,52] (Fig. 3a, b). These data support the hypothesis that talin-R in activated talin enhances the ability of talin-H to cooperate with kindlin-2 to promote potent integrin binding to multivalent ligands. How does talin-R achieve this enhancement? Since both kindlin-2 and tlnM3 bind to integrin-β CT in a nonexclusive manner (Supplementary Fig. 3a) with no detectable interaction between kindlin-2 and tlnM3 (Supplementary Fig. 3b), we wondered whether kindlin-2 and talin-R of tlnM3 are physically linked by a common binding partner, thereby further enhancing talin-dimer-induced microclustering of integrins to more efficiently bind to multivalent ligands. Since paxillin was shown to interact with talin[41,53,54] and kindlin[55–57] (see also Supplementary Fig. 3b), and all three proteins colocalize in nascent focal adhesions to regulate cell adhesion[55,58,59], we speculated that paxillin might physically link activated talin with kindlin-2 to regulate integrin activity. Remarkably, expression of a paxillin-binding defective kindlin-2 mutant G42K/L46E[56] (Fig. 3d) and knockdown of paxillin (Fig. 3e, f) both significantly reduced the synergistic action of tlnM3/kindlin-2 on integrin activation. Furthermore, co-expression of paxillin and tlnM3 resulted in a synergistic increase (~2.5-fold) in PAC-1 binding compared with tlnM3 alone (Fig. 3g) and such synergy was also significantly reduced when a kindlin-2-binding defective paxillin-F577E mutant[56] was used (Fig. 3g) or kindlin-2 was knocked down (Fig. 3h, i). By contrast, paxillin (also kindlin-2) had no significant effect on tlnM3-mediated monomeric FN10 binding to integrin-αIIbβ3 (Supplementary Fig. 3c). However, either paxillin or kindlin-2 cooperates with talin to induce potent oligomeric FN10 binding to integrin (Supplementary Fig. 3d), which is similar to the PAC-1-binding data in Fig. 3. Overall, these data provide strong evidence that paxillin links activated talin with kindlin-2 to promote potent integrin binding to multivalent ligands. We note that although a previous study had proposed that paxillin may bridge talin-H (instead of talin-R) and kindlin for promoting integrin activation[41], the study did not observe any synergistic effect of integrin activation when paxillin and talin-H were co-expressed vs talin-H alone. Supplementary Fig. 3e also shows that co-expression of paxillin and talin-H induced no synergistic effect on PAC-1 binding compared with the additive effect of expressing talin-H and paxillin separately. The slight enhancement of PAC-1 binding by the paxillin/talin-H co-expression vs talin-H alone was caused by paxillin that can induce a small effect of PAC-1 binding when expressed alone, see Supplementary Fig. 3e. This is in sharp contrast to the dramatic synergistic effect of co-expressing tlnM3 and paxillin (Fig. 3g), strongly supporting our hypothesis that paxillin acts through talin-R to promote potent integrin activation. A question is: if paxillin does not affect talin-H activity by linking talin-H and kindlin-2, how does overexpression of talin-H and kindlin-2 result in some synergy in the integrin binding to PAC-1[9–13,36,37] (also see Fig. 3a)? One possible mechanism is that overexpression of talin-H, which is known to recruit PIPKIγ to produce PIP2[23,24] that in turn binds and activates talin[15,17,18], may result in the activation of some endogenous talin, which then triggers

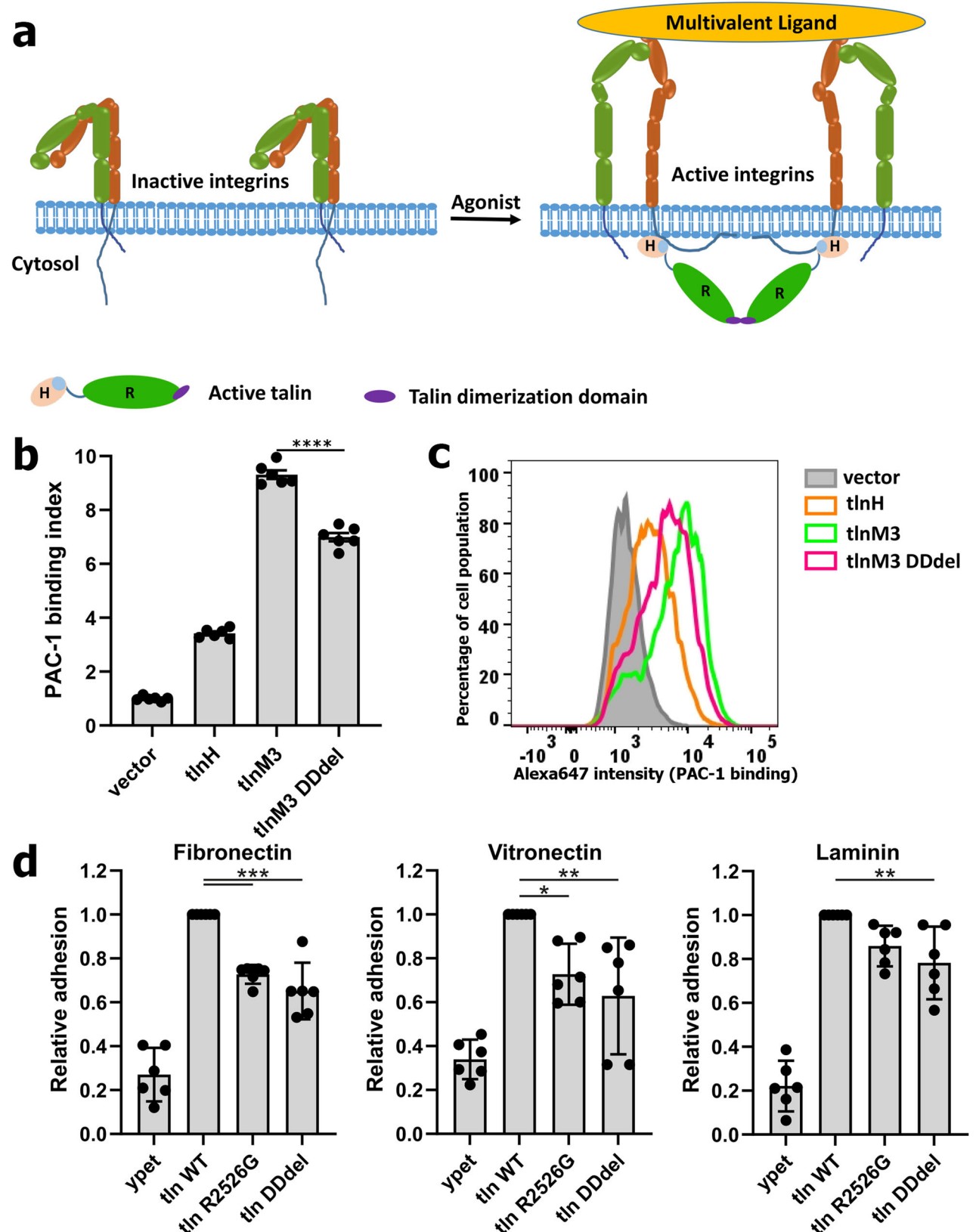

endogenous talin (via talin-R)/paxillin/kindlin pathway. Consistent with this possibility, the talin-H/kindlin-2 synergy to induce PAC-1 binding was significantly reduced when endogenous talin-1 was knocked down (Supplementary Figs. 3f and 3g), demonstrating that endogenous talin is indeed involved in

mediating the talin-H/kindlin-2 synergy. Furthermore, while paxillin has no effect in the talin-H activity regulation (Supplementary Fig. 3e), the talin-H/kindlin-2 synergy to induce the PAC-1 binding was significantly reduced by the paxillin-binding defective kindlin-2 mutant G42K/L46E (Supplementary Fig. 3h).

**Fig. 2 Dimerization of active talin is required to promote potent ligand binding to integrin. a** A model of multivalent ligand binding to integrin mediated by talin dimerization. **b** Integrin activation triggered by active full-length talin (tlnM3) is partially reduced by deletion of dimerization domain (tlnM3 DDdel) as measured by the PAC-1-binding assays. All values are given as mean ± S.E.M. ****$p < 0.0001$ with 95% confidence interval −2.814 to −1.826 ($t$-test), $N = 6$ biologically independent samples. **c** Histogram of PAC-1 binding shows that DDdel (magenta) causes significantly reduced PAC-1 staining. **d** Cell adhesion to fibronectin, vitronectin, and laminin are all affected when we introduce endogenous level of talin DD deletion or R2526G mutation to talin$^{1/2dko}$ fibroblasts to disrupt talin dimerization. WT values were set to 1. Relative cell adhesion was plotted as mean ±95% CI, $N = 6$ independent experiments. Fibronectin: ***$p < 0.0001$, Vitronectin: *$p = 0.0233$, **$p = 0.0018$, Laminin: **$p = 0.0097$. Statistical significance was tested by one-way ANOVA followed by Tukey's multiple-comparison test. **b**, **d** Raw data are provided in Source Data file.

These data provide evidence that activation of endogenous talin/paxillin/kindlin-2 pathway contributes to the known talin-H/kindlin-2 synergy in activating integrin in CHO-A5 system.

Given the potential indirect effect in the CHO system due to protein overexpression, we decided to underpin the role of the talin/paxillin/kindlin-2 complex on regulating integrin activity and function again by using the talin-1 and talin-2 double-deficient fibroblast system (talin$^{1/2dko}$). We retrovirally expressed ypet-tagged talin-H, tlnWT, and tlnM3 and sorted for cells with similar ypet levels (Supplementary Fig. 3i), which also showed comparable integrin-surface levels, and then analyzed cell adhesion on fibronectin and vitronectin, as well as cell spreading on fibronectin. As expected, talin-H was unable to fully promote cell adhesion and completely failed to induce cell spreading likely due to the absence of talin-R to connect with paxillin and kindlin as well as actin filaments (Fig. 4a, b). By contrast, however, tlnWT and tlnM3 expression induced very similar cell adhesion and spreading (Fig. 4a, b). We note that the retrovirally expressed tlnWT becomes activated by the endogenous activation machinery and does not require activating mutations, which is different from the CHO model. Next, we compared talin-H- and talin-WT-dependent cell adhesion in the presence of kindlin-2 and upon CRISPR/Cas9-mediated kindlin-2 gene deletion (Fig. 4c), which revealed that a synergistic effect of talin and kindlin-2 on cell adhesion can only be found in cells expressing full-length talin-WT but not talin-H (Fig. 4d). This observation is consistent with the above CHO cell-based data that the talin and kindlin-2 cooperativity on cell adhesion requires the presence of talin-R. We then reexpressed mCherry-tagged kindlin-2 WT and paxillin-binding deficient kindlin-2 G42K/L46E mutant in kindlin-2-deficient fibroblasts that express talin-WT, sorted cells with similar mCherry expression, and then analyzed cell adhesion on fibronectin-coated surfaces. In line with the above CHO data, expression of the paxillin-binding deficient kindlin-2 G42K/L46E mutant (Fig. 4e) showed significantly reduced cell adhesion compared with cells expressing kindlin-2 WT (Fig. 4f). Moreover, siRNA knockdown of paxillin also led to reduced cell adhesion in kindlin-2 WT-expressing cells but did not further reduce adhesion in kindlin-2 G42K/L46E- expressing cells (Fig. 4f). We also knocked out kindlin-2 in mouse embryonic fibroblasts (MEFs) and observed significantly reduced adhesion of cells expressing the paxillin-binding defective kindlin-2 G42K/L46E mutant compared with cells that were rescued with wild-type kindlin-2 (Supplementary Fig. 4a and 4b). Moreover, only expression of paxillin-WT but not the kindlin-binding deficient paxillin-F577E mutant in paxillin siRNA-mediated knockdown MEFs induced cell adhesion, indicating again the critical importance of the paxillin–kindin-2 interaction (Supplementary Fig. 4c and 4d). Overall, these data collectively showed that disruption of paxillin–kindlin interaction impairs ligand binding and cell adhesion, which is fully consistent with the above CHO cell-based data and also with the previously observed defects of focal adhesion formation, cell spreading, and migration induced by the disruption of this interaction[60].

**Molecular insight into the binding of paxillin to talin.** Paxillin is a multidomain adaptor containing five LD motifs at its N-terminus followed by four tandem double zinc-finger LIM domains[60] (Supplementary Fig. 5a). To further elucidate the mechanism of how paxillin binds to talin for regulating integrin activation, we undertook detailed structure-based mapping of the interaction. We first purified full-length paxillin, tlnM3, and talin-WT, and performed GST-based pull-down assays. Interestingly, purified paxillin exhibits stronger binding to tlnM3 than talin-WT in these pull-down assays (Fig. 5a). We next performed pull-down experiments using GST-paxillin and various talin fragments. Supplementary Fig. 5b shows that talin R1–R4 is a major fragment to bind to paxillin. Consistently, talin1–1047 containing only talin-H and talin R1–R4 (deleting the C-terminal 1048–2541 containing the dimerization domain and autoinhibitory talin-R9, see Fig. 1b) still activates integrin much more potently than talin-H (Supplementary Fig. 5c), consistent with the crucial role of talin-R1–R4 to mediate the talin/paxillin interaction in connection with kindlin. Among other talin-R fragments, talin-R9–R12 also displayed binding to paxillin, but this interaction was much weaker than talin-R1–R4 (Supplementary Fig. 5b). Paxillin also binds to talin-H but to a significantly lesser extent than talin-R1–R9 or talin-R1–R4 (Supplementary Figs. 5b and 5d). The dominant role of talin-R1–R4 in binding to paxillin was further confirmed by the quantitative SPR experiments (see below). On the paxillin side, deletion-based binding analysis revealed that paxillin N-terminal 1–160 containing LD1 and LD2 (Supplementary Fig. 5a) plays a major role in binding to talin (Fig. 5b and Supplementary Fig. 5d). Consistently, deletion of paxillin 1–160 almost completely abolished the paxillin synergy with tlnM3 to induce PAC-1 binding (Supplementary Fig. 5e). On the other hand, expression of paxillin 1–160, which does not contain the major kindlin-2-binding site located in paxillin LIM4 domain (Supplementary Fig. 5a) (see Zhu et al. 2019[56]), also hardly showed any synergistic effect on tlnM3-induced PAC-1 binding (Supplementary Fig. 5e). These data provide further evidence that full-length paxillin is required to act as a link between activated talin (primarily via paxillin N-terminal 1–160) and kindlin-2 (via paxillin C-terminal LIM4 domain) to form the talin/paxillin/kindlin axis and promote high-affinity integrin binding to multivalent ligands.

To further characterize the binding mode between talin and paxillin in more detail, we prepared individual $^{15}$N-labeled talin-H subdomains and talin-R1–R13 subdomains and examined their binding to paxillin 1–160 using NMR-based heteronuclear single quantum correlation (HSQC) experiments. The strength of the binding can be roughly estimated by the extent of the chemical shift changes with and without the binding partner. In talin-R, R1, R5, R6, R7, R9, R10, and R12 had no chemical shift changes (not shown), whereas R2, R3, R4, R8, R11, and R13 had chemical shift changes upon addition of paxillin 1–160. Among these paxillin-binding subdomains, R2, R8, and R11 showed significantly bigger chemical shift changes by paxillin (Fig. 5c, Supplementary Figs. 6a and 6b) than R3 (Supplementary Fig. 6c) and R4 (Supplementary Fig. 6d), respectively. R13 had very tiny/

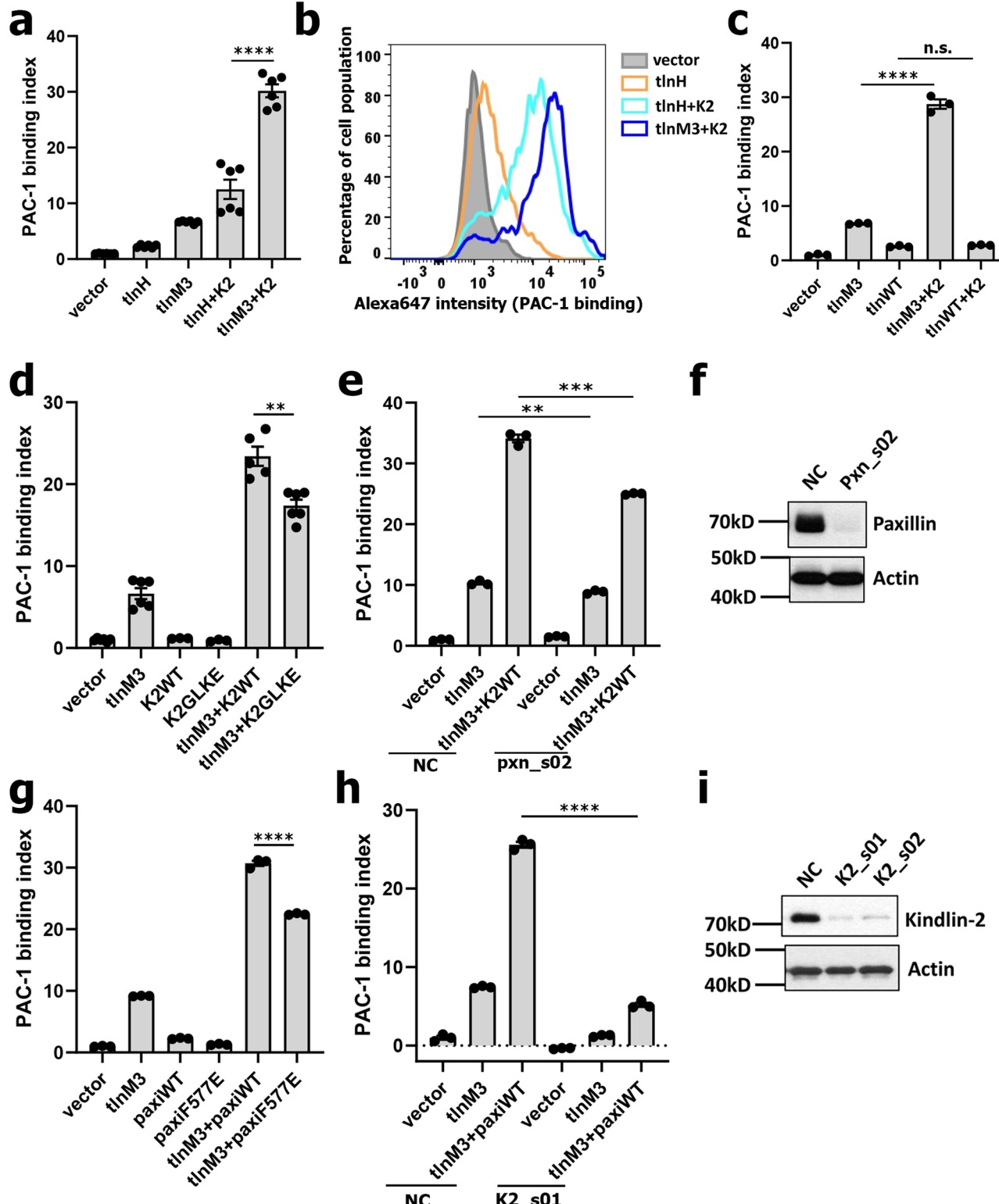

almost negligible chemical shift changes by paxillin (Supplementary Fig. 6e). In talin-H, only talin-F3 exhibited chemical shift changes upon addition of paxillin 1–160 (Supplementary Fig. 6f). Table S1 provides the summary of the HSQC-based binding results for all talin subdomains. Quantitative SPR experiments on those subdomains that showed significant chemical shift changes by paxillin 1–160 revealed that paxillin 1–160 binds talin-R2 at $K_D \sim 2.21\ \mu M$ (Supplementary Fig. 7a), to talin-R8 at $K_D \sim 5.37\ \mu M$ (Supplementary Fig. 7b), to R11 at $K_D \sim 12.7\ \mu M$

(Supplementary Fig. 7c), and to F3 at $K_D \sim 3.51\ \mu M$ (Supplementary Fig. 7d). As expected from the pull-down data (Supplementary Figs. 5b and 5d), talinR1–R4 as the dominant paxillin-binding core exhibited strong affinity to paxillin 1–160 at $K_D \sim 153\ nM$ (Supplementary Fig. 7e). It is important to note that talin-R1–R4 binding to paxillin 1–160 fits very well in 1:1 ratio by two-state model, suggesting that talin-R2, as the main binding module, may cooperate with R3 or R4 to bind paxillin via a multisite binding mode. As a comparison, talin-F3, talin-R8,

**Fig. 3 Kindlin cooperates with intact active talin via paxillin to activate integrin. a** Kindlin-2 synergizes with intact active talin (tlnM3 + K2) to trigger PAC-1 binding to integrin-αIIbβ3. This effect is drastically lower in cells expressing similar level of talin head and kindlin-2 (tlnH+K2). ****$p = 0.0001$ with 95% confidence interval 13.02–22.32 (t-test), N = 6 biologically independent samples. Values are shown as mean ± S.E.M. **b** Representative histogram of (**a**) where tlnM3 + K2 (dark blue) enhances PAC-1 binding much more drastically than tlnH+K2 (light blue). **c** Talin WT does not cooperate with kindlin-2 in PAC-1-binding tests while tlnM3 does. ****$p < 0.0001$ with confidence interval 19.58–24.34 (t-test), N = 3 biologically independent samples; ns, $p = 0.0815$ with confidence interval −0.0459 to 0.5080 (t-test), N = 3 biologically independent samples. Values are shown as mean ± S.E.M. **d** Kindlin cooperation with full-length active talin (tlnM3 + K2WT) was significantly impaired when the paxillin-binding defective kindlin-2 mutant G42K/L46E (tlnM3 + K2GLKE) was expressed. **$p = 0.0014$ with 95% confidence interval −9.039 to −3.018 (t-test), N = 6 biologically independent samples. Values are shown as mean ± S.E.M. **e** Paxillin knockdown substantially reduced the synergy of tlnM3/kindlin-2 to activate integrin. Note that the effect of tlnM3 alone on PAC-1 binding was also significantly reduced (~15%), albeit less than that by tlnM3/kindlin-2 co-expression (~25%) and the latter had higher effect probably due to the overexpression of both tlnM3 and kindlin-2 vs tlnM3 alone in the former. **$p = 0.0058$ with 95% confidence interval −2.228 to −0.7115 (t-test); ***$p = 0.0001$ with 95% confidence interval −10.86 to −7.379 (t-test), N = 3 biologically independent samples. Values are shown as mean ± S.E.M. **f** Western blot showing that paxillin was efficiently knocked down. One out of two independent experiments is shown here. **g** Paxillin synergizes with tlnM3 to induce potent integrin activation. The synergy (tlnM3+paxiWT) was impaired by kindlin-2-binding defective paxillin mutant F577E (tlnM3 + paxiF577E). ****$p = 0.0001$ with 95% confidence interval −9.442 to −7.087 (t-test), N = 3 biologically independent samples. Values are shown as mean ± S.E.M. **h** Kindlin-2 knockdown substantially reduces the synergy of tlnM3/paxillin to activate integrin. ****$p < 0.0001$ with 95% confidence interval −21.65 to −19.14 (t-test), N = 3 biologically independent samples. Values are shown as mean ± S.E.M. **i** Western blot showing that paxillin was efficiently knocked down. One out of two independent experiments is shown here. Uncropped images (**f**, **i**) and raw data (**a**, **c**–**e**, **g**, **h**) are provided as a Data Source file.

and talin-R11, of which each has an isolated binding site, binds much weaker than talin-R1–R4. Interestingly, despite the presence of multiple domains/sites within paxillin and talin, SPR experiments revealed that binding of full-length paxillin to tlnM3 also fitted well with a 1:1 model at $K_D \sim 33.1$ nM (Supplementary Fig. 7f), indicating that multiple sites in one paxillin may bind to multiple sites in one talin, resulting in 1:1 binding. This is reminiscent of talin/vinculin binding in 1:1 ratio, despite that there are 11 vinculin-binding sites on talin-R[18]. Nevertheless, talin-R1–R4 clearly contributes most significantly to the tlnM3 binding to paxillin.

The binding of paxillin to talin-F3 in the talin-H is interesting since such binding competes with the autoinhibitory R9 binding to talin-F3 (Supplementary Fig. 8a), providing a basis for understanding why paxillin preferably binds to activated talin (Fig. 5a). More interestingly, such binding competes with the talin-F3 binding to integrin-β CT when we used higher protein concentration in the HSQC experiments (Supplementary Fig. 8b), indicating that the paxillin- and integrin-binding sites on talin-F3 overlap. However, given the modest affinity of paxillin binding to talin-F3 ($K_D \sim 3.51$ μM, Supplementary Fig. 7f), strong interaction between integrin-β CT and talin-H in PIP2-enriched membrane media[61] (which is also reflected in our SPR tests where integrin-β CT was immobilized to CM5 chip to bind talin-H, Supplementary Fig. 1) would readily replace paxillin from talin-F3 when paxillin is recruited by talin bound to both membrane and integrin. In other words, paxillin may temporarily stabilize the activated talin by binding to talin-H, but once talin-H binds to integrin-β CT, paxillin is dissociated from talin-H. Indeed, paxillin had very little effect on talin-H-mediated integrin activation (Supplementary Fig. 3e). On the other hand, paxillin had drastic effect (Fig. 3g) on tlnM3 apparently via the talin-R/paxillin/kindlin-2 pathway in combination with the dimerization effect (see more detailed analysis below). Our data suggest that the engagement of talin-R with paxillin is crucial for spatial clustering of integrins by linking talin to kindlin in addition to talin dimerization, thereby leading to potent integrin binding to multivalent ligands. To further examine this hypothesis, we decided to perform more detailed structural mapping studies to identify specific paxillin sites on talin-R, so that we can perform more definitive mutation-based functional analysis.

Previous structural analysis indicated that paxillin LD motifs bind to talin-R8 that forms a helical bundle similar to other paxillin LD-binding modules[54]. We therefore synthesized paxillin LD1 and LD2 peptides (Supplementary Fig. 9a) that are present

in paxillin 1–160 fragment and examined their binding to talin R2, R8, and R11 using HSQC. The LD peptides induced similar chemical shift perturbation spectra, but less extensive chemical shift changes, on talin-R2, R8, and R11 as compared with paxillin 1–160. To illustrate this more clearly, we plotted residue-based chemical shift changes of talin R2 and R8 by LD peptides and paxillin 1–160 (Supplementary Fig. 9b and 9c, respectively) (BMRB codes for the chemical shift assignments of talin R2 and R8 are 17350 and 19339, respectively). The perturbed residues that are surface-exposed are highlighted on the structures of talin-R2 (Supplementary Fig. 9d) and talin-R8 (Supplementary Fig. 9e), respectively, which reveal potentially similar LD-binding site in these domains. To further elucidate the LD-binding mode, we built a HADDOCK-based structure model of talin-R2 bound to LD2 by using chemical shift changes and the conserved interface residues in LD motif as constraints (Supplementary Fig. 9b showed that LD2 binds more tightly to R2 than LD1, see details in the method section and Supplementary Table 3 for statistics). Figure 5d shows the best-calculated model where the helix in LD2 binds to a hydrophobic groove formed by helices α1 and α2 of talin R2. Important interfacial interactions are well conserved in talin-R2 as compared with those of known structure of talin-R8 (Supplementary Fig. 9f) bound to DLC1-LD peptide[54] (PDB code 5FZT) (Supplementary Fig. 9a), which includes the hydrogen bond between D146 of LD2 and K687 of R2, and a couple of hydrophobic packing pairs: LD2–L152/R2 A680 and A710, LD2–L148/R2 T707, and LD2–L145/R2-I703. Structure-based alignment of R8 with R11 shows that R11 can also use the conserved surface to bind LD motif (Supplementary Fig. 9g), as consistent with the NMR- (Supplementary Fig. 6b) and SPR- (Supplementary Fig. 7c) binding data. These analyses illustrated how LD motifs bind similarly to these helical bundle domains in talin-R. A key question is: given the 1:1 binding between paxillin and tlnM3 (Supplementary Fig. 7f), how do the LD motifs in paxillin engage with multiple talin subdomains via a multisite binding mode? To address this question, we focused on paxillin 1–160 (<18 kDa) and talin-R1–R4 (~70 kDa) for NMR-based binding studies since they play major role in the paxillin/talin interaction as indicated by the above studies. Supplementary Fig. 10a shows that $^1$H–$^{15}$N HSQC of paxillin 1–160 exhibits a narrow spectral window, indicating that it is largely unstructured. However, titration of unlabeled talin-R1–R4 induced chemical shift changes or line broadening of a dozen residues (Supplementary Fig. 10a). The perturbed residues mostly belong to LD1

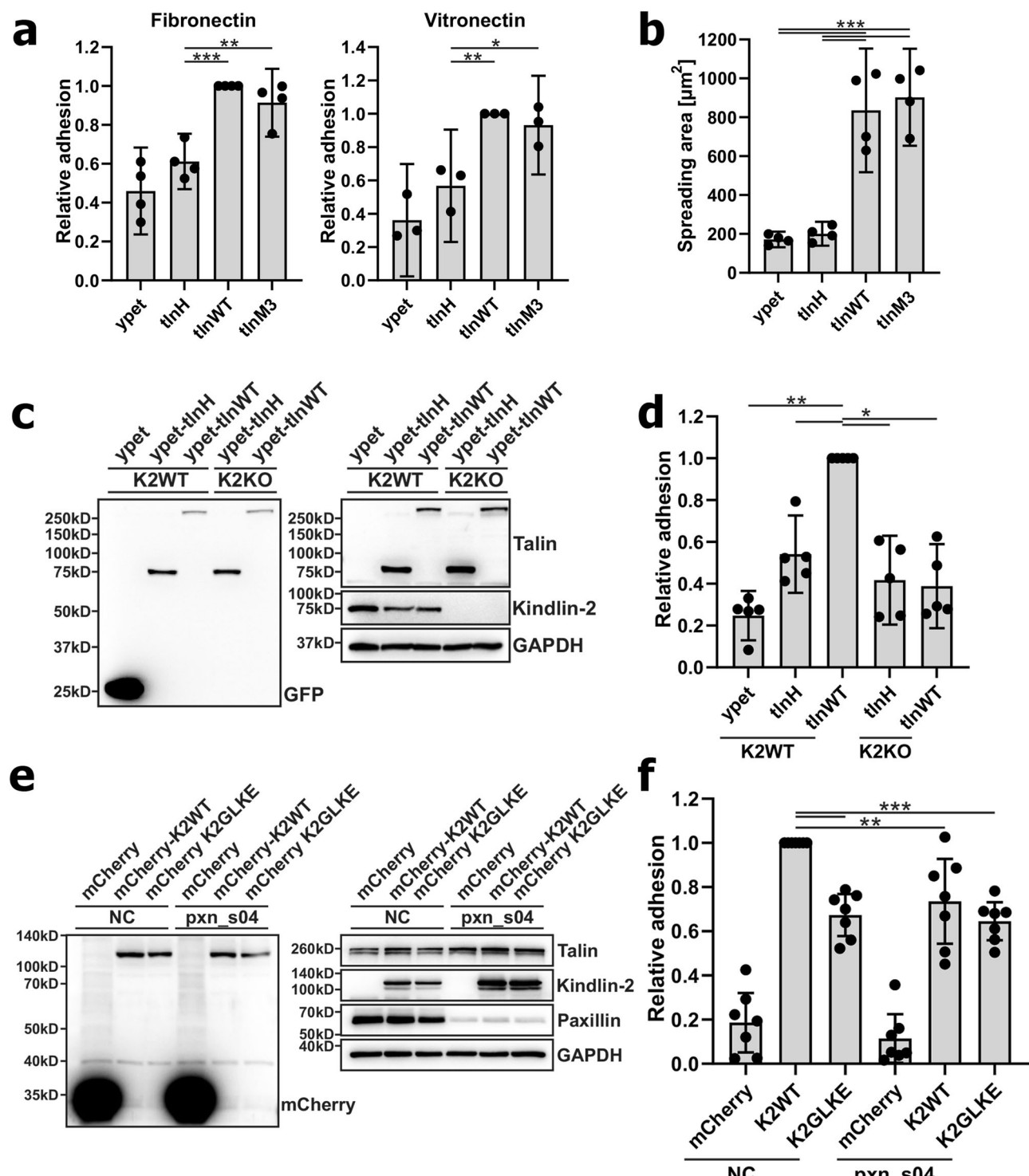

and LD2 (Supplementary Fig. 10a) since different set of perturbed signals disappeared upon deletion of N-terminal LD1 region (Supplementary Fig. 10b) or C-terminal LD2 region (Supplementary Fig. 10c). Representative peaks from LD1 and LD2 are selected to show their perturbations during talin-R1–R4 binding at four different ratios (Supplementary Fig. 10d). Remarkably, Alphafold2[62] was able to predict a model in which LD2, which binds tighter to R2 than LD1 (Supplementary Fig. 9b), chooses to bind talin-R2, whereas LD1 is bound to talin-R3 that was shown to weakly bind to paxillin 1–160 (Supplementary Fig. 6c and Table 1). Thus, it is possible that paxillin uses multiple LD motifs

and surrounding regions that are spaced within hundreds of residues to engage with multiple talin subdomains, leading to stronger binding. While the atomic details of such multisite binding of the entire paxillin/tlnM3 complex (~700 kDa as a dimer) remain to be determined, Supplementary Fig. 10e at least provided some insight into how such multisite binding may occur through the interaction between paxillin 1–160 and talin-R1–R4, which represents the core unit of the paxillin/tlnM3 complex as elucidated above.

Based on the conserved LD-binding surfaces in talin-R2 (Fig. 5d and Supplementary Fig. 9d), R8 (Supplementary Fig. 9e and 9f), and

**Fig. 4 Full-length talin WT cooperates with kindlin through paxillin to induce stronger cell adhesion in comparison with talin-H. a** Cell adhesion of talin$^{1/2dKO}$ fibroblasts expressing ypet, ypet-tagged talin head (tlnH), talin WT, or constitutively active talin (tlnM3) on fibronectin- or vitronectin-coated surfaces. WT values were set to 1. Data represent mean ±95% CI. $N = 4/3$ experiments. Fibronectin: **$p = 0.0050$, ***$p = 0.0007$, Vitronectin: *$p = 0.0179$, **$p = 0.0068$ (one-way ANOVA followed by Tukey's multiple-comparison test). **b** Cell-spreading area of cells expressing tlnH, tlnWT, or tlnM3 seeded on fibronectin after 30 min. Data are presented as mean ±95% CI. $N = 4$ experiments. ***$p < 0.0001$. (**c/d**) Cell adhesion of talin$^{1/2dKO}$ rescued with ypet alone, ypet–tlnH, or ypet–tlnWT in the presence (K2WT) or absence (K2KO) of kindlin-2. Kindlin-2 was knocked out by CRISPR/Cas9, K2WT cells were treated with a nontargeting control guideRNA. (**c**) Expression levels of ypet, full-length talin, talin head, and kindlin-2 assessed by Western blot. GAPDH served as loading control. **d** Static adhesion assays of the mentioned cell lines on fibronectin-coated surfaces. Data represent mean ±95% CI. $N = 5$ independent experiments. *$p = 0.0339$ (tlnH/K2WT vs tlnWT/K2WT), $p = 0.0276$ (tlnWT/K2WT vs tlnH/K2KO), $p = 0.0199$ (tlnWT/K2WT vs tlnWT/K2KO), **$p = 0.0072$ (one-way ANOVA followed by Tukey's multiple-comparison test). **e** Western blots showing expression of mCherry, talin, kindlin-2, and paxillin in K2KO ypet–talinWT-expressing cells retrovirally transduced with expression constructs carrying mCherry or mCherry-tagged wild-type (K2WT) or paxillin-binding defective (K2GLKE) kindlin-2 and subsequently treated either with negative control (NC) or paxillin-targeting (pxn_s04) siRNA. GAPDH served as loading control. Representative blots from one out of four experiments are shown. **f** Static adhesion assay of transduced cells with or without paxillin-targeting siRNA treatment. Data represent mean ±95% CI. $N = 7$ independent experiments. **$p = 0.0052$, ***$p = 0.0004$ (K2WT/NC vs K2GLKE/NC), $p = 0.0001$ (K2WT/NC vs K2GLKE/pxn_s04) (one-way ANOVA followed by Tukey's multiple-comparison test). Uncropped images (**c**, **e**) and raw data (**a**, **b**, **d**, **f**) are provided as a Data Source file.

R11 (Supplementary Fig. 9g), we designed LD-binding defective mutations in talin-R2, R8, and R11 as talin-R2 A680E/K687E (AKEE) (Fig. 5e), R8 A1537E/K1544E (AKEE) (Supplementary Fig. 11a), and R11 T2126E/K2133E (TKEE) (Supplementary Fig. 11b). We confirmed that all these mutations drastically reduced the binding of these subdomains to paxillin 1–160 as summarized in Fig. 5f, Supplementary Fig. 11c and 11d, respectively. We then generated these specific paxillin-binding defective mutants in tlnM3, tlnM3–R2-mut (A680E/K687E, AKEE), tlnM3–R2-R11mut (A680E/K687E/T2126E/K2133E, AKEE/TKEE), and tlnM3–R2–R8–R11mut (A680E/K687E/A1537E/K1544E/T2126E/K2133E) (illustrated in Fig. 6a) to examine the effect of the talin/paxillin cooperativity in inducing integrin activation. The combined mutations from talin-R2, R8, and R11 had the strongest effect when co-expressed with paxillin-WT (Fig. 6b). Remarkably, when testing tlnM3–R2–R8–R11 mutant with a kindlin-2-binding defective paxillin mutant (F577E)[56], the PAC-1 binding was further reduced to almost the level of tlnM3 alone (Fig. 6b). These data provide strong structure-based evidence that paxillin binding to talin-R is crucial for promoting integrin activation via a talin/paxillin/kindlin-2 complex.

To further support the relevance of the talin/paxillin/kindlin-2 complex for integrin activation and function in cells, we again used the talin$^{1/2dko}$ fibroblast system[19,21]. These cells were retrovirally transduced with constructs encoding for C-terminally ypet-tagged talin fusion proteins, which carried the above-mentioned mutations in the talin R2 (AKEE) and the talin R11 domains (TKEE), as well as combined mutations (AKEE/TKEE). Cells expressing identical ypet levels that match endogenous talin protein levels of control cells (talin1$^{fl/fl}$/talin2$^{-/-}$) were sorted and used for the experiments (Supplementary Fig. 12a). Moreover, all cell groups showed comparable kindlin-2, paxillin, as well as integrin-surface levels as confirmed by Western blotting and flow cytometry (Supplementary Fig. 12a and 12b). We then tested whether the mutations within the talin R2 and R11 domains affect paxillin binding in the cell by performing anti-GFP-immunoprecipitation experiments. Consistent with the GST-pulldown and NMR experiments, both mutants showed reduced binding to paxillin with the talin R2 AKEE mutation being most disruptive (Fig. 6c). Importantly, these talin mutations also significantly reduced the amount of coprecipitated kindlin-2 (Fig. 6c), providing strong cell-based evidence that talin is connected to kindlin through paxillin. However, such ternary complex, although detectable by Co-IP, was too dynamic to be co-eluted directly with size-exclusion column. It is conceivable that within cells, the talin/paxillin/kindlin-2 complex is further strengthened by interactions with the plasma membrane, as both talin and kindlin-2 interact with membrane lipids[42,43,63]. The size of the ternary complex (~800 kDa as mediated by dimeric tlnM3,

~540 kDa) is also too large to be characterized by NMR. However, considering that kindlin-2 F0 represents the major domain in binding to paxillin at C-terminal LIM4[60], we examined $^{15}$N-labeled kindlin-2 F0 binding to paxillin in the absence and presence of talin-R1–R4 (binds to paxillin at 1–160). Supplementary Fig. 12c shows that while addition of paxillin to $^{15}$N-labeled kindlin-2 F0 caused line broadening of many residues in kindlin-2 F0 due to its interaction with paxillin (~70 kDa), addition of talin-R–R4 (~70 kDa) caused additional line broadening of many perturbed residues in kindlin-2 F0 apparently due to the talin-R1–R4-binding (to paxillin) induced size increase, thus providing strong evidence for the ternary complex formation. Next, we investigated whether a disruption of the talin/paxillin/kindlin-2 complex affects cell adhesion, focal adhesion formation, and cell spreading, which are initiated upon the integrin/ligand interaction. To this end, we first plated the cells on the multivalent integrin ligands commonly existing in the extracellular matrix (fibronectin, vitronectin, and laminin) and found that while talin$^{1/2dko}$ cells expressing ypet as a control that hardly adhered to the substrata, the R2 and R2/R11 double-mutant cells showed an average 30–40% reduction in cell adhesion, whereas the R11 mutant showed a smaller adhesion defect compared with cells expressing WT talin (Fig. 6d). We then plated the cells on FN and measured cell spreading for 4 h. While ypet-transduced talin$^{1/2dko}$ cells remained roundish and did not spread over time, in comparison with WT talin, both paxillin-binding mutants displayed a significant spreading defect, which became even more prominent, when both sites were mutated (Fig. 6e). Consistent with our adhesion data, the R2 mutant revealed a more pronounced spreading defect compared with the R11 mutant (Fig. 6e). Next, we characterized the adhesion structures of the different cell lines by immunofluorescence staining (Supplementary Fig. 12d). Interestingly, while we observed a similar focal adhesion (FA) size between the groups, their number and the relative FA area relative to the total cell area were also strongly reduced (Fig. 6f). To further test whether impaired paxillin binding affects the localization of talin to focal adhesions, we measured the intensity of ypet fluorescence within FAs in relation to the total cellular ypet fluorescence intensity. Our measurements showed significantly reduced localization of paxillin-binding defective talin in FAs, suggesting that paxillin–talin interaction is crucial for recruiting/maintaining talin within FAs (Fig. 6g and Supplementary Fig. 12d). Moreover, we also detected significantly less kindlin-2 in FAs (Fig. 6g and Supplementary Fig. 12d), further supporting the hypothesis that talin via paxillin maintains/stabilizes kindlin-2 within FAs or vice versa. In addition, we found that the number of active β1 integrins within FAs (indicated by 9EG7 staining) was significantly reduced in cells expressing the talin R2 AKEE mutant (Fig. 6g). These data provide

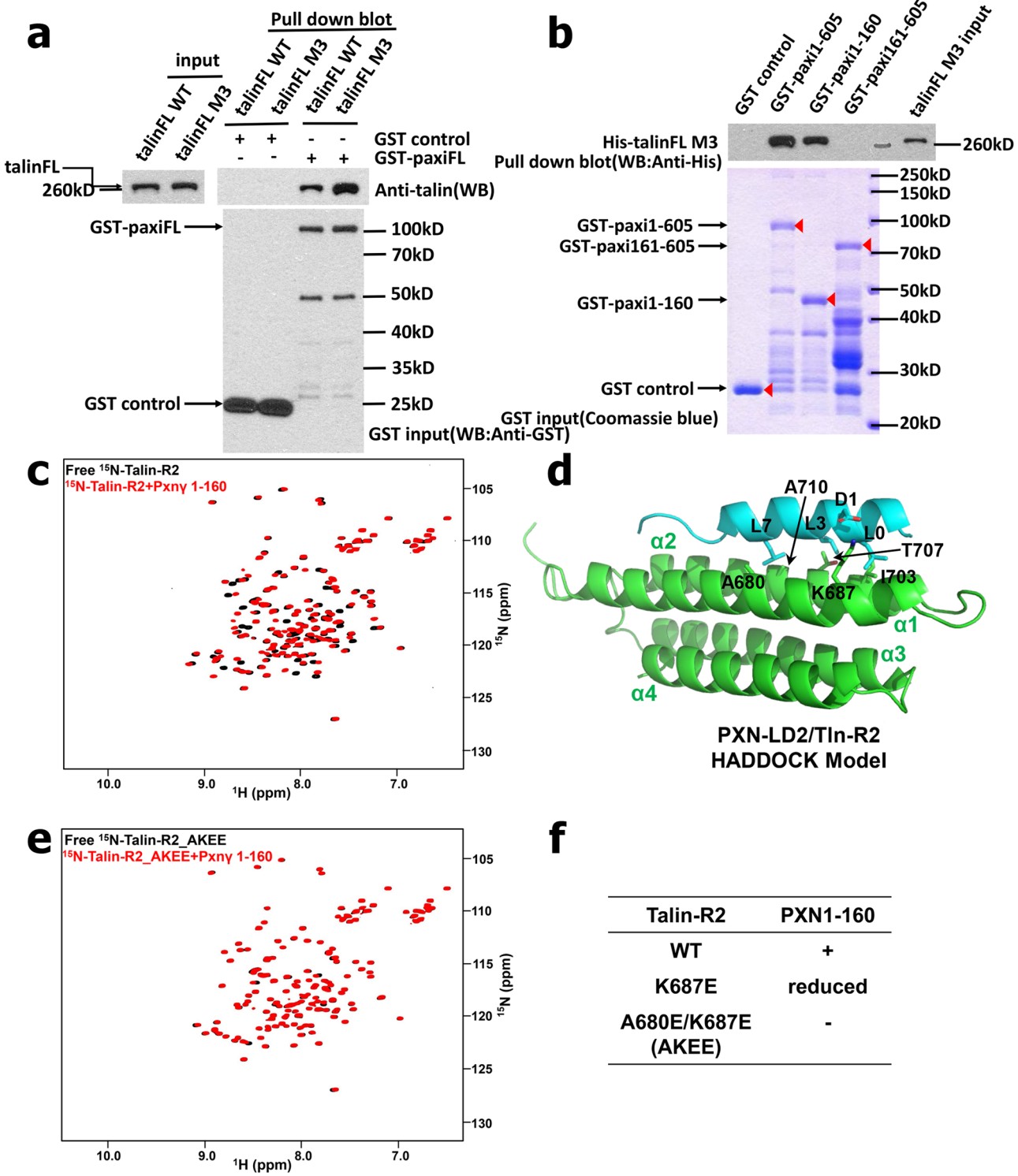

strong evidence that paxillin binding stabilizes talin as well as kindlin-2 within focal adhesion, and thus promotes integrin activation. Overall, a disruption in the talin–paxillin interaction causes reduced talin/paxillin/kindlin complex in FAs, resulting in reduced integrin activation and impaired cell adhesion and spreading.

**A talin-mediated supramolecular machinery orchestrates potent integrin activation.** The above data identified two critical

talin-R-dependent pathways to promote talin-H-induced integrin activation: (i) talin-R-mediated talin dimerization and (ii) talin-R-mediated talin–paxillin–kindlin-2 complexation. How do these two pathways cooperate? We show that disruption of the talin dimerization significantly reduced the ability of tlnM3 to activate integrin (Fig. 7a), but disruption of both dimerization and paxillin binding led to further significant reduction in integrin activity (Fig. 7a and Supplementary Fig. 13a). Furthermore, the synergistic effect of tlnM3 and kindlin-2 on integrin activity was also reduced by the disruption of the talin dimerization, and even

**Fig. 5 Paxillin binds to full-length active talin through multiple sites. a** Paxillin prefers binding to activated talin as shown by GST pulldown. GST-tagged full-length paxillin was immobilized to GST beads and incubated with the same amount of full-length wild-type talin (talinFL WT) and full- length activated talin (talinFL TM). Pull-down fraction of talin was detected by talin antibody on western blot. Immobilized GST proteins were detected by GST antibody on western blot after 10x dilution. Four independent experiments were performed. **b** N-terminal part of paxillin (paxi1–160) is mainly responsible for binding to full- length active talin, while paxillin C-terminal (paxi161–605) shows no binding to talinFL TM. Pull-down fraction of talin was detected by Anti-6xHis antibody on western blot. Immobilized GST protein was resolved by SDS-PAGE and shown by Coomassie blue staining. As can be recognized from Coomassie staining, GST-paxillin 161–605 became unstable upon the 1–160 deletion, which degraded further after gel filtration. To alleviate this problem, we loaded much more paxillin 161–605 to have the major band (see arrow) to match other paxillin inputs, yet there was still no talin pulled down, indicating that paxillin 161–605 does not contribute to talin binding in contrast to paxillin 1–160. Four independent experiments were performed. **c** The HSQC spectra of 50 μM $^{15}$N-labeled talin-R2 in the absence (black) and presence (red) of 100 μM unlabeled paxillin-γ 1–160 (red) showing direct interaction between paxillin and talin R2. **d** Cartoon representation of the best model of Paxillin-LD2 (in cyan) and talin-R2 (in green) complex through Haddock calculation. Side chains of critical residues involved in the binding are displayed and labeled. Red-dashed lines indicate potential H-bonds. **e** Mutations in talin-R2 (A680E/K687E, AKEE) abolished talin-R2 binding to paxillin 1–160 at the same experimental condition as (**c**). **f** Summary of mutation testing results from HSQC. Uncropped images are provided in Source Data file.

more substantially reduced when both pathways were disrupted (Fig. 7b, Supplementary Fig. 13b and 13c). These data clearly demonstrate the existence of a supramolecular machinery involving dimerized talin and talin/paxillin/kindlin-2 linkage to promote potent integrin activation and cell adhesion.

## Discussion

Unlike many transmembrane receptors such as GPCRs, ion transporters, cytokine receptors, and receptor tyrosine kinases, which are activated by binding to extracellular ligands, the ECM ligand binding of integrins can be induced by intracellular cues[7]. Such unique inside-out activation process prevents spontaneous or unbalanced ligand binding to integrin, allowing tight control of many adhesion-dependent physiological responses such as platelet aggregation, leukocyte adhesion, muscle contraction, and tissue regeneration. While talin-H is well known to induce integrin-α/β CT separation that in turn triggers conformational change of the ectodomain, which is widely thought as the final step of integrin activation (Fig. 1b), our studies here revealed for the first time that talin-R also plays a crucial role in this process by dramatically enhancing the talin-H activity. Mechanistically, we showed that talin-R connects active talin dimer (via paxillin) to kindlin to form a vital integrin-activation complex to promote microclustering of integrins to potently bind to multivalent ligands that are characteristic of natural integrin ligands. Talin-R thus has a dual role in regulating the integrin activity in a spatiotemporal manner: (a) it acts as a negative regulator by masking the integrin-binding site on talin-H in the autoinhibited state of talin to prevent integrin activation; (b) upon the conformational activation of talin by PIP2[15,17,18], it then acts as a positive regulator by forming a vital membrane-associated integrin- activation complex. Our detailed molecular mapping studies revealed that N- and C-termini of paxillin bind talin and kindlin, respectively, providing a basis for understanding how paxillin gets recruited to integrin-containing nascent adhesions by engaging with both talin[64] and kindlin[53–56]. Based on these analyses, we propose a significantly updated model of integrin-activation process (Fig. 7c): upon talin activation, talin-H of the activated talin disrupts the integrin-α/β CT interaction, triggering an inside-out conformational change of the receptor[25,26], which allows its initial contacts with ligands as indicated by its binding to monovalent fibronectin FN10 repeat. Meanwhile, the activated talin, which is dimerized via talin-R, clusters two talin-H-activated integrin molecules. Such clustering is further strengthened by the talin/paxillin/kindlin complex, thereby leading to preassembled active integrin microcluster to potently bind to multivalent ligand (Fig. 7c). This model is strongly supported by our extensive functional data in both CHO cells and fibroblasts that are associated with β1 and β3 integrins. Although further

studies need to be performed, other integrins likely function via the same mechanism since talin, paxillin, and kindlin are broadly expressed. It is conceptually important to emphasize here that the talin dimer/paxillin/kindlin complex-induced integrin micro-clustering (Fig. 7c) is part of the integrin inside-out signaling process that triggers integrin high-affinity and stable integrin-ligand interactions, but is fundamentally different from post-ligand-induced cell-adhesion events such as focal adhesion formation, cell spreading, and migration that are associated with integrin macroclustering (avidity regulation). Such microclusters of activated integrins are advantageous for forming multiple contacts with ligands leading to an immediate increase in integrin-ligand-binding strength, which may be particularly important for the rapid formation of firm adhesions, such as those required for leukocyte-endothelial interactions to withstand the shear forces of flowing blood. The integrin microclusters connected by the talin/paxillin/kindlin complex also provide a key platform for transitioning integrin inside-out activation to outside-in signaling, resulting in the assembly of nascent adhesion complexes that interact with other adapter and signaling proteins, such as ILK, FAK, and actin filaments to ensure growth and assembly of the large focal adhesion complex (Fig. 7c). How this complex process is regulated remains to be determined, but may depend on the stoichiometry of talin, kindlin, and paxillin in different cell types. Paxillin has been shown to be one of the first proteins present at sites where focal adhesions form and talin and kindlins are expressed at similar levels in various cell types[38] and so variations in paxillin- expression levels or its local enrichment at the membrane may be critical for regulating the formation of the complex and focal adhesion assembly. Additional mechanisms may exist to fine-tune the talin dimer/paxillin/kindlin-mediated integrin microcluster formation and subsequent focal adhesion formation. For example, several recent studies observed a small degree of kindlin oligomerization under certain experimental conditions such as low-salt concentration. Such kindlin oligomerization was thought to also contribute to the talin-H/kindlin synergy[42,43,63]. However, other studies reported (a) only monomeric form for kindlin-2/kindlin-3 in buffers mimicking physiological conditions[65–68], and (b) oligomerization of kindlin inhibits integrin activation[38]. Thus, the role of kindlin oligo-merization in general and in the context of the talin/paxillin/kindlin complex remains to be further investigated. It is important to emphasize that integrin microclustering in our model (Fig. 7c) is facilitated by talin and kindlin binding to the β-CTs of two different integrins. While talin and kindlin may bind to the β-CT of the same integrin in a nonexclusive manner (Supplementary Fig. 3a), it remains to be further investigated how such binding affects activation of single-integrin molecule as co-expression of tlnM3 and K2 had only a small effect on the

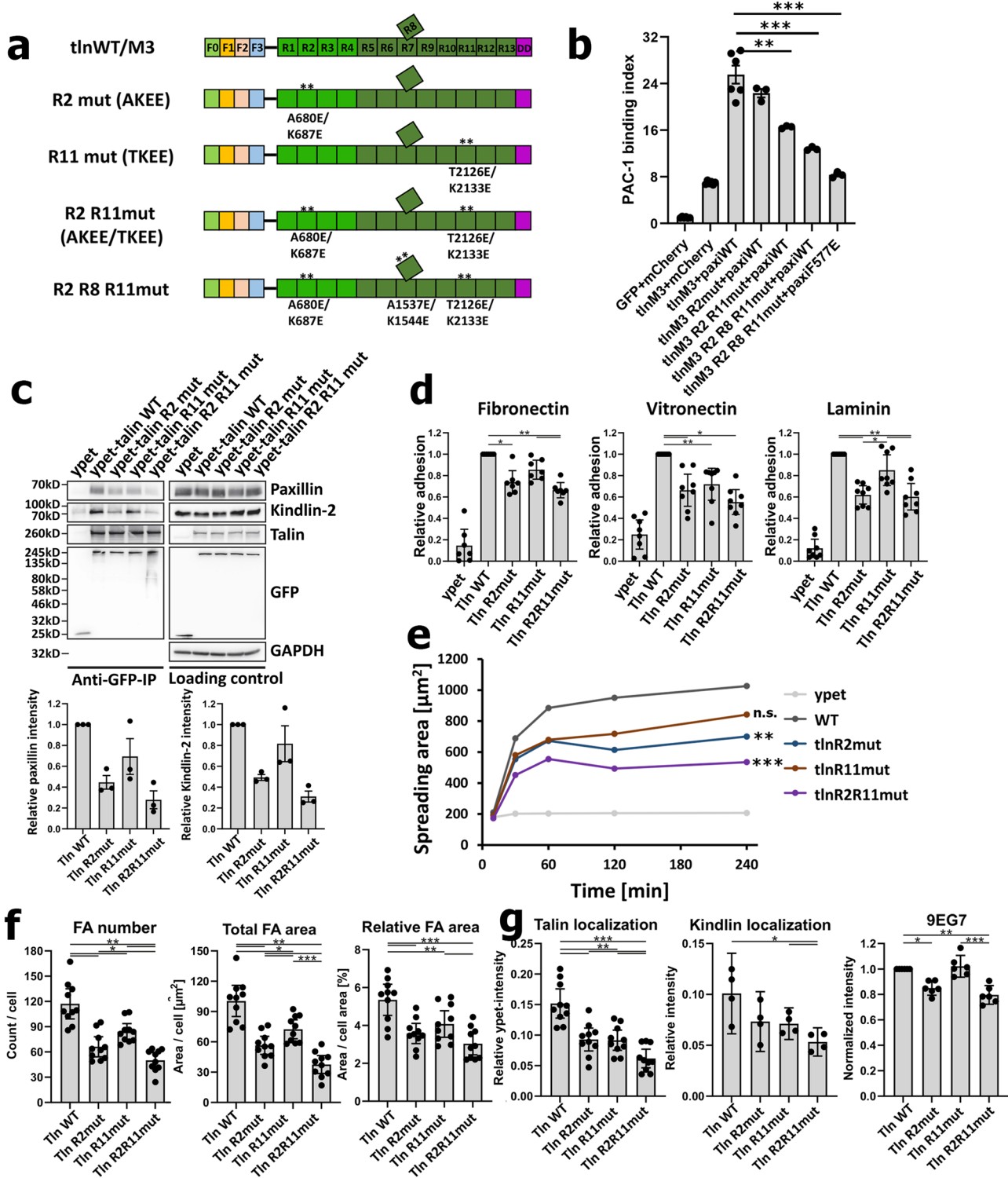

integrin binding to monomeric ligand compared with tlnM3 alone (Supplementary Fig. 3c). It also remains to be determined if any weak talin/kindlin interaction exists beyond the detection limit of common methods, or is strengthened by unknown cellular factors such as post-translational modifications, which may regulate the integrin microclustering. However, a recent single-molecule imaging study found that talin and kindlin-2 are actually closely spaced within ~20-nm distance in association with active β1 integrins[69], supporting our finding that the two proteins

do not directly associate with each other, but are bridged by the mediator paxillin to cluster different integrins.

In summary, our studies significantly advance the understanding of integrin activation through spatiotemporal regulation of talin by paxillin that further links talin dimer with kindlin into a supramolecular complex. We have found that talin-R is extremely important for such paxillin-mediated talin/kindlin linkage. The architecture of this talin/paxillin/kindlin complex machinery is not only important for promoting potent integrin activation

**Fig. 6 Paxillin-binding defective mutations in talin impair integrin activation and cell adhesion. a** Diagram of paxillin-binding defective talin mutants. **b** Paxillin-binding defective mutations on talin rod (R2mut/ R2R11mut/R2R8R11mut) reduced the tlnM3/paxillin synergy (tlnM3+paxiWT) to activate integrin. Combined paxillin-binding defective mutations on talin rod (tlnM3 R2R8R11mut) and kindlin-binding defective mutation on paxillin C-terminus (paxiF577E) lead to the lowest PAC-1 binding, comparable to that of only expressing active talin (tlnM3). **$p = 0.0054$ with 95% confidence interval $-14.37$ to $-3.650$ (t-test), $N = 3$ biologically independent samples; ***$p = 0.0008$ with 95% confidence interval $-18.10$ to $-7.368$ (t-test) or 0.0001 with 95% confidence interval $-22.60$ to $-11.85$ (t-test), $N = 3$ biologically independent samples. Values are given as mean ± S.E.M. **c** Immunoprecipitation experiments (talin[1/2dKO] fibroblasts expressing ypet–talin WT/paxillin- binding mutants) show that talin R2 (AKEE) and R11 (TKEE) mutations both reduced paxillin binding and co-immunoprecipitation of kindlin-2. Relative paxillin and kindlin-2 intensities were plotted as mean ± S.E.M. Paxillin and kindlin-2 intensities were normalized to intensities of immunoprecipitated talin, $N = 3$ independent experiments. **d** Cell adhesion to fibronectin, vitronectin, and laminin are all affected by talin R2 (AKEE) and R11 (TKEE) mutations. WT values were set to 1. Relative cell adhesion was plotted as mean ± 95% CI, $N = 8$ experiments. Fibronectin: *$p = 0.0198$, **$p = 0.0036$ (WT vs R2R11mut), $p = 0.0073$ (R11mut vs R2R11mut), Vitronectin: *$p = 0.0205$ (WT vs R2mut), $p = 0.016$ (WT vs R2R11mut), **$p = 0.0048$, Laminin: *$p = 0.0278$, **$p = 0.0045$ (WT vs R2mut), $p = 0.0035$ (WT vs R2R11mut), $p = 0.0013$ (R11mut vs R2R11mut) (one-way ANOVA followed by Tukey's multiple-comparison test). **e** Both R2 (AKEE) and R11 (TKEE) mutations lead to reduced cell spreading. Spreading areas of 30 cells per cell line and time point were measured per experiment, $N = 6$. **$p = 0.0022$, ***$p = 0.0002$ (one-way ANOVA followed by Sidak's multiple-comparison test). **f** Both R2 (AKEE) and R11 (TKEE) mutations reduced focal adhesion (FA) number as well as total and relative FA area per cell, but only a combination of both rod mutations reduced FA size. FAs were characterized by quantification of paxillin immunofluorescence stainings shown in Fig. S12D. Data are shown as mean ±95% CI, $N = 10$ experiments (8 cells were analyzed per group and experiment). **g** Localization of talin and kindlin-2 to FAs was reduced in cells expressing paxillin-binding defective talin mutants. Active β1-integrin-specific antibody 9EG7 staining intensities within FAs were reduced in immunofluorescence stainings of cells expressing R2 and R2/R11 mutant talin variants. Bar plots represent mean ±95% CI, $N = 10$ or 4 or 6 experiments (8 cells were analyzed per group and experiment). **f**, **g** *$p < 0.05$, **$p < 0.01$, ***$p < 0.001$. Statistical significance was tested by one-way ANOVA followed by Tukey's multiple-comparison test. Uncropped images (**c**), exact p-values (**f–g**), and raw data (**b**, **d–g**) are provided in Source Data file.

---

but also for triggering rapid post-ligand cell-adhesion events such as focal adhesion assembly, signaling, cell spreading, and migration. Future studies are needed to further investigate how this machinery is regulated under physiological and pathological conditions. Given the critical role of integrin activation in these responses, our findings may be highly valuable for guiding future development of diagnostics and therapeutics to treat many integrin-dependent diseases.

## Methods

**Plasmid constructs and mutagenesis**. The following constructs for bacterial expression were used in this study: mouse full-length talin subcloned in a pET28T vector, human full-length paxillin-γ subcloned in a pGST-1 vector, and mouse kindlin-2 subcloned in a pET28-SUMO vector. Talin subdomains are subcloned into pHis-1 vector or pET28T vector according to solubility and digestion sites. Paxillin fragments are subcloned into pGST-1 vector. Fibronectin repeat 10 (1541–1639, FN10) was subcloned into pET28a and pGST-1 vectors, respectively. The following constructs for mammalian expression were used in this study: human full-length paxillin-γ (1–605) subcloned into pmCherry-c1 vector and human full-length kindlin-2 (1–680) subcloned into pmCherry-c1 vector. Mouse full-length talin (1–2541), talin head (1–433), and talin rod (434–2541) subcloned into pEGFP-c1 vector were obtained originally from Addgene. Construct mutagenesis was conducted by using QuikChange lightning site-directed or Multi site-directed mutagenesis Kit (Agilent Technologies). Mutated constructs generated in this study include tlnM3 (M319A, T1767L, and E1770K), talinR2mut (A680E, K687E), talinR8mut (A1537E, K1544E), talin R11mut (T2126E, K2133E), and talinDDdel (Q2483stop). Note that pET28T is a modified pET28a vector where the thrombin-cleavage site is substituted with a TEV protease cleavage site. Primers used in this study are listed in Supplementary Table 4.

**Protein expression and purification**. Recombinant proteins with fusion tag (6xHis tag or GST tag) were expressed in *Escherichia coli* BL21 (DE3) strain (New England Biolabs). Typically, bacteria were initially grown in 50 ml of Lysogeny broth (LB) medium and then amplified in 2 L of LB at 37 °C. The culture was induced by 0.4 mM isopropyl β-D-1-thiogalactopyranoside (IPTG) at 20 °C or room temperature (RT) for overnight when it reached an A600 of 0.6. The pellet was then collected and suspended in buffer and frozen at −80 °C. For protein purification, the pellet was lysed by incubation with lysozyme and sonication. After high-speed centrifugation, the supernatant was subjected to affinity-column purification by using either nickel or GST gravity column. Gel filtration was always performed in the final step by using either Superdex-75 (16/60) or Superdex-200 (16/60, GE Healthcare). Note that Superose-6 (10/300) was used for full-length talin purification. Purified protein was checked by sodium dodecyl sulfate poly-acrylamide gel electrophoresis (SDS-PAGE). The fusion tag (either His tag or GST tag) was removed by TEV protease during protein purification when applied for NMR analysis. Isotope-labeled ([15]N) proteins were achieved by employing minimal

medium with [15]NH$_4$Cl. Protein concentration was measured by absorbance at 280 nm or Pierce BCA protein assay kit (Thermo Fisher Scientific).

His-FN10 and GST-FN10 were purified using affinity columns and GST tag was removed by TEV protease. Both His-FN10 and GST-cleaved FN10 were further purified by size-exclusion column, which both eluted at the monomer position (<17kD according to protein marker on Superdex-75 increase 10/300 GL). Purified proteins were then subject to biotin labeling in DPBS using EZ-link Sulfo-NHS-Biotin (Catalog#21217, Fisher) on ice for 2 h. Biotin-labeled FN10s were purified again through FPLC (Superdex-75 increase 10/300 GL) and the spectrum was overlaid with the FPLC profile before biotin labeling. Interestingly, we found that after biotin labeling, the His-FN10 (His-FN10-biotin) was oligomerized, eluting at ~600-kD position, while FN10 (without tag) with biotin labeling (FN10-biotin) was still eluted at <17kD corresponding to monomer. This biotin-induced oligomer of His-FN10 was then used as multivalent ligand control for mono-FN10 in our ligand-binding assay to better characterize the talin–paxillin–kindlin-induced integrin activation.

**NMR experiments**. 2D-HSQC experiments were performed on Bruker 600-MHz NMR spectrometer with Bruker NMR Topspin software (version 4.0.6) at 25 °C (except for the titration experiments shown in Supplementary Fig. 12c, 5 °C was applied to retain homogeneous solution). Samples were generally made of 50 μM [15]N-labeled protein and the desired ratio of unlabeled binding partner proteins or peptides in a buffer containing 50 mM NaH$_2$PO4/Na$_2$HPO4 (pH 6.8), 50 mM NaCl, and 2 mM DTT (1,4-dithiothreitol) or 0.5 mM TCEP (tris(2-carboxyethyl) phosphine) in the case of full-length paxillin. In total, 1024 and 256 data points were collected in proton and nitrogen dimensions, respectively. In total, 32 scans were used to achieve sufficient signal-to-noise ratio. DSS was used to reference [1]H chemical shifts and indirectly reference [15]N. Standard sine window function, zero filling, and Fourier transformation were used to process the data, and the spectra were visualized via NMRPipe[70] (version 9.2) and NMRFAM-SPARKY[71] (version 3). Amide nitrogen and proton chemical shifts of residues in talin-R2 and talin-R8 were assigned according to the published BMRB database with access code 17350 and 19339, respectively. Chemical shift change ($\Delta\delta_{obs}$ [HN,N]) was calculated with the equation $\Delta\delta_{obs}$ [HN,N] $= [(\Delta\delta\ HN\ W\ HN)^2 + (\Delta\delta\ N\ W\ N)^2]^{1/2}$, where W HN and W N are weighting factors based on the gyromagnetic ratios of [1]H and [15]N (W HN = 1 and W N = 0.154) and $\Delta\delta$ (p.p.m.) = δ bound − δ free.

**Docking simulations**. Models of the talin R2 and paxillin-LD2 complex were calculated from molecular docking simulations using the HADDOCK webserver (version 2.4, wenmr.science.uu.nl/haddock2.4/)[72,73]. Talin R2 (655–782) structure extracted from talin R1R2 crystal structure (PDB code: 1SJ8) and paxillin-LD2 (141–159) structure extracted from the crystal structure of human-paxillin-LD2 motif in complex with FAK fragment (PDB code: 4XGZ), were used as starting structures. Docking restraints were derived from the chemical shift perturbations observed during paxillin 1–160 NMR titration. "Active" residues, which are most perturbed and solvent-accessible residues that likely take place in the binding interface, included A674, S677, A681, K687, S688, Q691, A704, T707, Q708, A710, L711, and S714 in talin R2. The classic interacting LD motif residues L145, D146, L148, and L152 were considered "active" in paxillin LD2. "Passive" residues, which

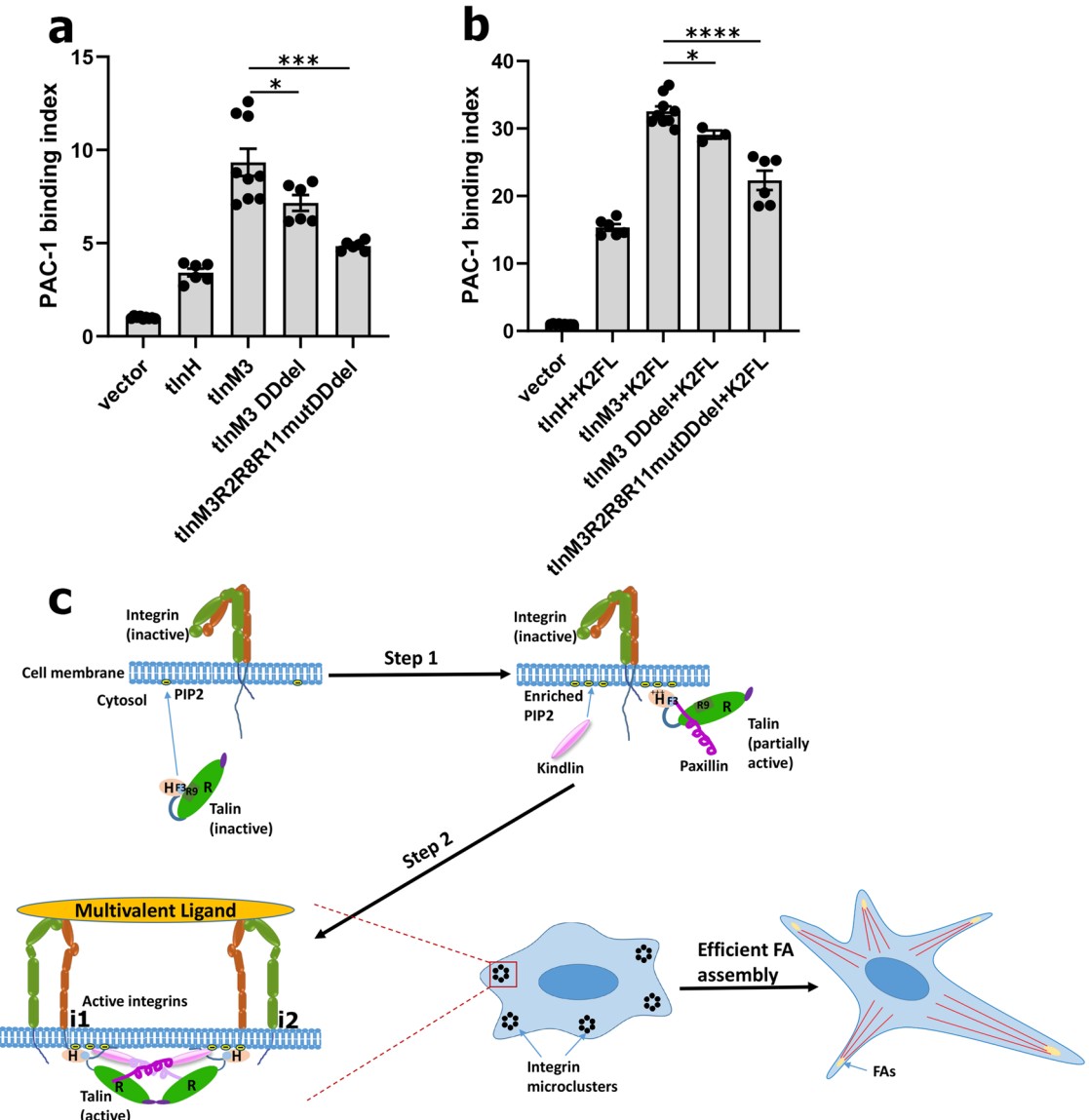

**Fig. 7 Both talin dimerization and talin–paxillin–kindlin pathway contribute to integrin activation. a** PAC-1-binding assay showing that integrin activation by full-length active talin (tlnM3) was reduced by deletion of talin-dimerization domain (tlnM3 DDdel), and addition of paxillin-binding mutations (tlnM3 R2R8R11mut) further reduced PAC-1 binding to almost the level caused by talin head (tlnH) only. *$p = 0.0416$ with 95% confidence interval −4.267 to −0.0963 (t-test); ***$p = 0.0003$ with 95% confidence interval −6.453 to −2.529; $N = 6$ biologically independent samples. Values are shown as mean ± S.E.M. **b** Synergy between intact kindlin-2 and intact active talin (tlnM3 + K2FL) was significantly impaired by DDdel as shown by reduced PAC-1 binding. Combination of paxillin-binding deficient mutations and DDdel mutation further substantially reduced PAC-1 binding. *$p = 0.0286$ with 95% confidence interval −6.485 to −0.4448 (t-test), $N = 3$ biologically independent samples; ****$p < 0.0001$ with 95% confidence interval −13.40 to −7.084 (t-test), $N = 6$ biologically independent samples. Values are given as mean ± S.E.M. **c** A model for dynamic integrin-activation process where inactive talin first engages with PIP2 to initiate, via talin-H, the conformational opening of talin, which initiates the recruitment of paxillin (Step 1). At Step 2, activated talin dimer, while anchored to PIP2 membrane, clusters two conformationally open integrins (**i1** and **i2**) via each talin monomer subunit. Meanwhile, talin, which is bound to integrin **i1**, links, via paxillin, to kindlin that is bound to another integrin **i2**. Such talin–paxillin–kindlin linkage (primarily driven by talin-R as shown in the text) strengthens the talin-dimer-induced microclustering of integrins, leading to their potent multivalent binding and efficient FA assembly for cell adhesion. **a**, **b** Raw data are provided in Source Data file.

are solvent-exposed neighbors to "active" residues, were defined automatically by HADDOCK server. Ambiguous interaction restraints (AIRs) were defined between the active residues of talin R2 and paxillin LD2. Eleven clusters with at least 4 structures in each cluster were generated and included total 147 structures out of the final 200 calculated from HADDOCK server, which represents 73% of the water-refined models. The statistics of each cluster is summarized in Supplementary Table 3. Overlaying all the models in the first two clusters that have the best Haddock score and contain the most structures shows a converged ensemble with heavy atoms RMSD 0.31 ± 0.11 Å and 0.26 ± 0.10 Å, respectively. Examining the representative model in each of these two clusters shows that paxillin-LD2 structures are similar but with opposite direction with respect to talin R2 domain. This is likely due to the highly symmetrically positioned Leu residues in the sequence of

LD motif (Supplementary Fig. 9a), and the docking program cannot specify the interacting residue groups without unambiguous distance constraints. Examining the remaining clusters reveals complex structures more or less similar to cluster 1 or 2.

**AlphaFold2 structure prediction**. The model of talin R1–R4 and paxillin 1–160 complex was generated via the publicly available Google ColabFold notebook (github.com/sokrypton/ColabFold)[62] that was modified from the Deepmind's original notebook[74]. The input includes the primary protein sequence and multiple-sequence alignment from MMseqs2[75] without any templates. Amber-relaxation option is selected to optimize side-chain bond geometry[76]. Structures

were visualized in PyMOL (version 2.4.1) [Schrodinger, L. L. C. The PyMOL Molecular Graphics System (2010)].

**GST pulldown.** A volume of ~25 µg of purified GST or GST-fused protein was immobilized on 15 µl glutathione-Sepharose 4B resin (beads) via incubation on a rotator for 1.5 h at 4 °C in the binding buffer containing 25 mM Tris-HCl (pH 7.5), 150 mM NaCl, 0.5 mM TCEP, and 0.01 or 0.1% NP40 (nonyl phenoxypolyethoxylethanol), and supplemented with Complete EDTA-free Protease Inhibitor (Roche). The desired amount of prey protein was then added in and incubated with beads for another 2 h at 4 °C. After that, the beads were washed by 500 µl binding buffer three times and subjected to denaturation by adding 30 µl 2 × SDS loading buffer and boiling for 5 min. After centrifuging at high speed, supernatants were resolved by SDS-PAGE or western blotting. All the pull-down experiments were performed at least twice independently.

**Western blotting and antibodies.** Western blotting was performed using standard protocol. Briefly, samples resolved by SDS-PAGE were transferred onto 0.45-µm polyvinylidene fluoride (PVDF) membrane (MilliporeSigma). The membrane was blocked with 5% nonfat milk in TBST buffer 1 h at RT. After that, the membrane was incubated with primary antibody at 4 °C overnight and then incubated with horeseradish peroxidase (HRP)-conjugated secondary antibody at RT for 1 h. The blots were detected by Pierce ECL Western Blotting Substrate (Thermo Fisher Scientific). The following primary antibodies were used in this study: GAPDH (D16H11) Rabbit mAb (Catalog# 5174, Cell Signaling Technology), Anti-Actin (Ab-1) mouse mAb (Catalog# CP01, MilliporeSigma), GST Tag Antibody (8-326) (Catalog# MA4-004, Thermo Fisher Scientific), Rabbit-anti human-paxillin (H-114) (Catalog# sc-5574, Santa cruz), Mouse-anti human-paxillin (D-9) (Catalog# sc-365174, Santa cruz), Anti-kindlin-2 Antibody (clone 3A3) (Catalog# MAB2617, MilliporeSigma), Anti-GFP (G5.1) XP antibody (Catalog# 2956, Cell Signaling Technology), Anti-talin antibody clone 8d4 (Catalog#T3287, MilliporeSigma), and Anti-talin antibody [1A11] (Catalog#ab57758, Abcam). Secondary antibodies included Anti-rabbit IgG, HRP-linked Antibody (Catalog# 7074 S, Cell Signaling Technology), and anti-mouse IgG, HRP-linked Antibody (Catalog# 7076 S, Cell Signaling Technology). Primary antibodies were used at 1:1000 dilution or 1:500 (paxillin and talin 8d4 antibody) dilution, and secondary antibodies were used at 1:3000 dilution.

**Surface plasmon resonance (SPR).** SPR studies were performed using BIAcore S200 (GE Healthcare) in 50 mM sodium phosphate (pH 6.8), 150 mM NaCl, 0.05% Tween 20, 0.03% BSA, and 0.05% NaN$_3$. GST or GST-integrin beta3CT or paxillin 1–160 or tlnM3 was covalently coupled to series S CM5 sensor chips via amine coupling where 10 µg/mL purified proteins in 10 mM sodium acetate (pH 4.5 or 5.0) were injected over the activated chip surface to reach a general density of 400 response units (RU) (tlnM3 is immobilized 1000 or 10000RU for different tests). A series of concentrations of talin subdomains (0, 750 nM, 1.5 µM, 3 µM, 6 µM, and 12 µM) or talinH or tlnM3 (0, 1.96 nM, 3.91 nM, 7.82 nM, 15.63 nM, 31.25 nM, 62.5 nM, and 125 nM) or paxiFL (0, 7.8 nM, 15.6 nM, 31.2 nM, 62.5 nM, 125 nM, 250 nM, and 500 nM) were prepared and injected at 20 µL/min over the surface where paxillin or integrin or talin was immobilized. All experiments were performed at 25 °C. Each test was repeated. Data were analyzed using BIAEvaluation software 1.1 (GE Healthcare) with 1:1 fitting or bivalent fitting or steady-state model (1:1 fitting was always applied first, if it is not fitted well, we try other models for better fitting to best correlate with other biochemical data). All the sensorgram traces were subtracted by the reference-channel response and the zero-concentration response during data analysis.

**Size-exclusion analysis.** In all, 100ul purified protein samples were injected into Superose-6 10/300 GL balanced with 25 mM Tris-HCl (pH 7.5) and 150 mM NaCl, and running at the flow rate of 0.5 mL/min. Elution fractions were collected as 0.5 mL per tube after 0.2CV after injection and subject to SDS-PAGE analysis.

**Integrin-activation assay.** CHO-A5 cells stably expressing integrin αIIbβ3 were transfected with pEGFP-c1 constructs (talin) together with mCherry-c1 constructs (paxillin or kindlin-2) using PEI (polyethylenimine). In the case of protein knockdown, cells were pretreated with siRNA purchased from Sigma (universal negative control#1 (NC, Catalog#SIC001) or siRNA duplexes (Catalog#VC30002) specifically designed for CHO cells: pxn_s02 CACUUUGCUCUGCACCCA-CUdTdT; K2_s01 GAACUUGCUGACUAUAUUAAdTdT; TLN_s03 CAG-CAAUUGACAGGACACUCAdTdT using lipofectamine RNAiMAX (Invitrogen) for 48 h and then transfected with jetOPTIMUS (Polyplus). About 24 h post transfection, cells were detached with PBS-based cell-dissociation buffer (Catalog#13151014, Gibco) and stained with 1:100 anti-αIIbβ3 activation-specific mAb PAC-1 (Catalog# 340535, BD Biosciences) and 1:800 anti-integrin beta3, PSI domain(AP-3) antibody (Catalog#EBW106, Kerafast) at RT for 30 min in HBSS buffer (Catalog#14025076, Gibco) containing 0.1% BSA. After washing and 2% formaldehyde fixation, cells were stained with 1:800 Alexa Fluor 647 Goat Anti-Mouse IgM (Catalog# 115-607-020, Jackson Immunology Laboratories) and 1:100 Rat anti-mouse IgG1 (Catalog#550083, BD) at 2 °C for 30 min. After washing, cells were subject to analysis using BD LSRFortessa Flow Cytometer with BD

FACSDiva™ Software (version 9.0). Live and positively transfected cells were gated using FlowJo software (version 10) for beta3 (AP-3) and PAC-1-binding analysis (diagram shown in Supplementary Fig. 14). WT vs mutant protein levels were comparable based on the fluorescence gating of GFP or mCherry tags. Knockdown efficiency was validated at least twice before and after overexpression by western blot. Values for PAC-1 binding were reported as median fluorescent intensities and the data were normalized to vector control and integrin-expression level detected by AP-3. Integrin-activation levels triggered by protein expression are presented by the means ± S.E.M. For monomeric FN10-biotin and multimeric His-FN10-biotin binding to cells, the proteins were incubated with cells at a concentration of 10 µg/ 100 µl at RT for 30 min and followed by 1:200 APC-streptavidin (Catalog#554067, BD Bioscience) staining.

Statistical significance for integrin-activation data was generated by unpaired t-tests using PRISM8 (GraphPad software).

**Generation of cell lines and cell culture conditions.** Murine talin1 and -2 double-knockout fibroblasts (talin[1/2dko])[19–22] were retrovirally transduced with pLPCX expression construct containing YPET alone, C-terminally YPET-tagged talin-1 cDNA (talin WT), YPET-tagged talin head only, YPET-tagged paxillin-binding deficient talin-1 mutants (R2 mut (A680E/K687E), R11mut (T2126E/ K2133E), and R2R11mut), or YPET-tagged talin-1 dimerization mutants (DDdel (dimerization domain deleted) and R2526G point mutant), or a YPET-tagged constitutively active talin mutant (talinM3, M319A/T1767L/E1770K). All pLPCX expression constructs were cloned by Gibson assembly using NEBuilder HiFi DNA Assembly Master Mix (New England Biolabs Inc.) according to the manufacturer's instructions. Sequences of used primers are listed in Supplementary Table 4. For production of viral particles, HEK293 cells were cotransfected with the respective pLPCX vector and the packaging plasmid pCl-Eco using Lipofectamine 2000 (Thermo Fisher Scientific). Virus containing HEK cell supernatants was supplemented with 5 µg/ml polybrene and transferred to talin[1/2dKO] cells in five infection cycles.

Kindlin-2 was knocked out using the CRISPR/Cas9 system in talin[1/2dko] fibroblasts rescued with talin H and talin WT expression constructs. Cells were electroporated with RNP complexes by the Neon Transfection system (Thermo Fisher Scientific) following the manufacturer's protocol. RNP complexes were allowed to form by co-incubation of single-guide RNAs (sgRNAs, purchased from Integrated DNA Technologies: Alt-R® CRISPR–Cas9 Negative Control, Catalog# 1072544 and kindlin-2 targeting sgRNA: CUGCUACGCGGACGGGACGU) with TrueCut Cas9 protein (Thermo Fisher Scientific). To enrich the electroporated cell pool for kindlin-2-deficient cells, cells were repeatedly seeded on collagen- coated surfaces and loosely adherent cells collected after 1 h. These kindlin-2-knockout cells were retrovirally transduced with pLPCX expression constructs carrying mCherry, mCherry-tagged wild-type kindlin-2 cDNA (K2WT), or a mCherry-tagged paxillin-binding defective kindlin-2 mutant (K2GLKE, G42K/L46E) as described for talin-expression constructs. Cells were FACS-sorted using a FACSAria™ II cell sorter (BD Biosciences, Heidelberg, Germany) based on their YPET and/or mCherry intensity to isolate cells with comparable talin (YPET) and kindlin-2 (mCherry) expression levels. Paxillin knockdown was performed using siRNA duplexes purchased from Sigma-Aldrich. Cells were transfected either with nontargeting control siRNA or paxillin-targeting siRNA (pxn_s04, sense sequence: CACUUUGUGUGCACCCACU[dT][dT]) using Lipofectamine RNAiMAX Reagent (Thermo Fisher Scientific) following the manufacturer's instructions and analyzed after 1–2 days.

Cells were lysed and characterized by western blotting using mouse anti-GFP (homemade supernatant from a mouse hybridoma cell line[19,77], generated at the Max-Planck-Institute of Biochemistry, Munich, Germany), mouse anti-talin (Catalog# T3287, Sigma-Aldrich), rabbit anti-talin (H-300, sc-15336, Santa cruz), mouse anti-paxillin (Catalog# 610051, BD Transduction Laboratories), mouse anti-kindlin-2 (Catalog# MAB2617, Merck), mouse anti-GAPDH (Catalog# CB1001-500UG, Millipore), goat anti-rabbit-HRP (Catalog#111-035-144, Jackson ImmunoResearch Laboratories), and goat anti-mouse-HRP (Catalog# 115-035-003, Jackson ImmunoResearch Laboratories Inc.) antibodies following standard protocols. Mouse anti-talin, mouse anti-GAPDH, goat anti-rabbit-HRP, and goat anti-mouse HRP antibodies were used at a dilution of 1:20,000. Mouse anti-paxillin, rabbit anti-talin, and mouse anti-kindlin-2 antibodies were used at a dilution of 1:5000, 1:1000, and 1:2000, respectively. Cells were cultured under standard conditions in DMEM GlutaMAX (Thermo Fisher Scientific) supplemented with 10% fetal bovine serum, 100 U/ml penicillin, 100 µg/ml streptomycin, and nonessential amino acids (all from Thermo Fisher Scientific).

Mouse embryonic fibroblast (MEF) cell line (ATCC#CRL-1503) and the MEF-derived kindlin-2-knockout cell line (clone1–3, kindly provided by Dr. Huan Liu[56]) were maintained in the DMEM supplemented with 10% fetal bovine serum. Cells were transfected with Lipofectamine RNAiMAX Reagent (Thermo Fisher Scientific) for siRNA knockdown and followed with jetOPTIMUS (Polyplus) for mCherry-tagged proteins.

**GFP immunoprecipitation from cell lysates.** Talin[1/2dKO] cells expressing YPET (GFP derivative) alone, or YPET-tagged talin WT or paxillin-binding mutants were used for GFP-immunoprecipitation experiments. Cells were trypsinized and allowed to readhere on fibronectin-coated surfaces for 4 h. Adherent cells were

washed with PBS and subsequently treated with 0.5 mM DSP (dithiobis(succini-midyl propionate), Thermo Fisher Scientific) dissolved in PBS for 10 min to cross-link proteins. The reaction was stopped by adding 50 mM Tris (pH 7.5) for 10 min and washing with PBS. The cells were lysed in MPER buffer (Thermo Fisher Scientific) containing protease and phosphatase inhibitors (Roche Diagnostics). About 1 mg of protein per sample was used for immunoprecipitation experiments using the μMACS GFP Isolation Kit (Miltenyi) following the manufacturer's instructions. Recovered immunoprecipitated protein and loading control samples were analyzed by SDS-PAGE and Western blotting.

**Adhesion and spreading assays**. For adhesion assays, polystyrol flat-bottom 96-well microplates (Greiner Bio-One, Frickenhausen, Germany) were coated with 10 μg/ml laminin (Sigma-Aldrich), 5 μg/ml fibronectin (Sigma-Aldrich), or 5 μg/ml vitronectin (STEMCELL Technologies, Köln, Germany) in coating buffer (20 mM Tris-HCl, pH 9.0, 150 mM NaCl, and 2 mM MgCl2) overnight at 4 °C. After blocking with 3% bovine serum albumin in PBS for 30 min at room temperature, $4 \times 10^4$ cells in DMEM containing 0.1% FBS and 25 mM HEPES were seeded per well, incubated for 30 min (20 min for MEF paxillin-knockdown rescue), and washed with PBS. Adherent cells were fixed with 4% paraformaldehyde (PFA) for 15 min and stained with 5 mg/ml crystal violet in 2% ethanol for 30 min. After washing, the remaining crystal violet was dissolved in 2% SDS and quantified by measuring absorbance at 595 nm using a microplate reader (Tecan, Männedorf, Switzerland). In the case of MEF adhesion, stained cell images were taken with 4x lens under bright field on inverted microscope with LAS X software (version 3.7.4) and cell numbers were quantified by ImageJ software. All experiments were per-formed in quadruplets.

For spreading analysis, cells were plated on fibronectin (5 μg/ml)-coated dishes and phase-contrast pictures were taken using an EVOSTM FL Auto life cell microscope (Thermo Fisher Scientific) at indicated time points after seeding the cells. Cell-spreading area of 30 cells per group at each time point was measured using ImageJ software (US National Institutes of Health). Statistical significance for adhesion and spreading data was tested by One-way ANOVA followed by Tukey's or Sidak's multiple- comparison test using PRISM9 (GraphPad software).

**Flow cytometry**. Integrin-surface expression was determined by FACS using a Cytoflex LX flow cytometer (Beckman Coulter). Cells were stained and analyzed in PBS supplemented with 2% FBS and 2 mM EDTA. Cells were incubated with Alexa Fluor 647-conjugated anti-integrin β1 (Catalog# 562153), anti-integrin β3 (Cata-log# 563523), anti-integrin α5 (Catalog# 564312, all three BD Biosciences), or anti-integrin α6 (Catalog# 17-0495-82, Thermo Fisher Scientific) antibodies or with biotinylated anti-integrin αV antibody (Catalog# 104103, Biolegend) and subse-quently with Cy5-labeled streptavidin (Catalog# 016-170-084, Jackson Immu-noResearch Laboratories). All antibodies were used at a dilution of 1:200. Data were analyzed using FlowJo software (version 10).

**Immunostaining and FA analysis**. Cells were cultured on fibronectin-coated glass coverslips (Thermo Fisher Scientific) for 4 h and fixed for 10 min with 4% PFA. For active β1-integrin staining, cells were incubated for 30 min with rat anti-active β1-integrin antibody (clone 9EG7; Catalog# 553715, BD Biosciences) diluted 1:100 on ice prior to fixation. The following antibodies were used for further staining fol-lowing standard protocols: Mouse anti-paxillin (Catalog# 610051, BD Transduc-tion Laboratories), rabbit anti-paxillin (Catalog# ab32084, abcam), rabbit anti-total β1 integrin (described in Azimifar et al. 2012[78]), and mouse anti-kindlin-2 (Cat-alog# MAB2617, Merck) as primary antibodies. Goat anti-mouse-Alexa Fluor 546 and goat anti-rabbit Alexa Fluor 647 (Catalog# A11003 and A21244, Thermo Fisher Scientific) as secondary antibodies. All antibodies were used at a dilution of 1:300, except rabbit anti-total β1 integrin (1:5000). Phalloidin–Alexa Fluor 647 was used at a dilution of 1:300 (Catalog# A22287, Thermo Fisher Scientific). Images were acquired using a Leica TCS SP8 X confocal microscope (Leica Microsystems, Wetzlar, Germany) equipped with 63x (numerical aperture 1.40) oil objective lens and Leica Confocal Software (LAS AF, version 3.0). All pictures were processed with ImageJ software (version 1.53k). FA area was defined by paxillin staining and quantified using ImageJ software. For quantitative analysis of talin recruitment to FAs, the total ypet signal intensity within the FA area was normalized to the total ypet fluorescence of the whole cell. Statistical significance was tested by One-way ANOVA followed by Tukey's multiple-comparison test using PRISM9 software.

**Reporting summary**. Further information on research design is available in the Nature Research Reporting Summary linked to this article.

## Data availability

Source Data are provided with this paper. HADDOCK docking structure (for talinR2/ paxillin-LD2 complex) is available in ModelArchive (modelarchive.org) with the accession code ma-fu8t5. AlphaFold prediction structure (for talinR1–R4/paxillin 1–160 complex) is available in ModelArchive (modelarchive.org) with the accession code ma-25vb5. Referenced structures are available in Protein Data Bank with code 4F7G, 5FZT, 1SJ8 and 4XGZ. Referenced chemical shift assignments are available in BMRB chemical

shift statistics with code 17350 and 19339. All data supporting the findings of this study are available from the corresponding authors qinj@ccf.org and m.moser@tum.de upon reasonable request.

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

## Acknowledgements

We thank technical assistance by Yinghua Chen. The work was supported by NIH grant R01 HL58758 to JQ, P01 HL73311, and P01 HL154811 to EP and JQ, the Deutsche Forschungsgemeinschaft (SFB914 TP A01) and BMBF (01GM1912, eRARE-LADOMICs consortium) to M.M, and NIH NMR instrumentation grant S10OD023432.

## Author contributions

F.L., L.Z., T.B., J.Y., Q.Y., and J.L. performed the experimental studies. E.F.P. was involved in data interpretation. F.L., L.Z., T.B., J.Y., E.F.P., Q.J., and M.M were involved in writing the paper. Q.J. and M.M. supervised the work and finalized the paper.

## Competing interests

The authors declare no competing interests.
