## [Peer Review File · Nature Communications]

REVIEWER COMMENTS

Reviewer #1 (Remarks to the Author):

The manuscript reports the discovery of a new insight of integrin activation process, where the c-terminal rod domain of protein talin (talin-R) can dimerizing activated talin to enhance the talin-H potency, via the help of an adaptor protein paxillin and integrin co-activator kindlin-2. The dimerization induces clustering of integrin, which is stabilized by the talin-paxillin-kindlin complex, leading to preassembled active integrin micro-cluster that can bind to multivalent ligand. The study is well designed and comprehensive with the utilization of many biophysics, biochemistry, and molecular cell biology tools. The manuscript is well written with sufficient details on the methods. The new model based on the extensive supporting data is convincing and dramatically advanced our understanding on integrin activation process, which is essential for cell growth, signaling spreading and migration.

I only have a couple questions/suggestions below:

For the SPR experiments, in the method section it is mentioned that each test was repeated, but I do not see the replicated data. How many times each test/concentration was repeated? The repeated response curves should be provided, and the mean and standard deviation (error range) of kinetic results based on the repeated measurements should be calculated. If the measurement were not repeated, please remove the repeat statement in method, and report the standard deviation of the fitting results (how well is the data fitting to the model).

Also for the SPR experiments, the paxillin^{FL} was measured at 10 degree C (Fig. S5F), while the rest were measured at room temperature. The binding kinetics is a function of temperature based on thermodynamic principle. Have the authors take this temperature variation into consideration? The weaker KD of 350 nM between Talin-TM/paxillin could be due to this lower temperature measurement. I suggest adding this point to the discussion in page 13.

Reviewer #2 (Remarks to the Author):

This manuscript examines how talins and kindlins work together for integrin activation. Currently most studies have shown that integrin activation only requires the talin head domain and has only used truncated talin head domain constructs. In this manuscript the authors, utilizing more physiologically relevant, full-length talin constructs, detail binding mechanisms between the talin rod domain, kindlin, paxillin and the subsequent activation of integrin α IIB β 3. The authors make a significant contribution to elucidating the critical first steps of focal adhesion formation and it is a significant addition to the literature on how integrins are activated. There are however some issues that need to be addressed. Experimentally the major issue is the use of an artificial α IIB β 3 integrin system in a CHO cell. This system is useful but also has some major problems especially with respect to understanding physiological relevance. These authors have talin double null cells so the manuscript would be significantly improved if they were used for more of the physiological readouts. In addition, the manuscript would benefit from significant revisions to extrapolate their findings more fully. Further, the details provided explaining the critical first steps of focal adhesion formation are arguably the most significant contribution of this paper and could be showcased more efficiently.

Major comments that would significantly improve the study include:

In figure 1 what happens to cell adhesion, spreading and integrin activation of beta 1 integrin in the double null fibroblasts with the TIn TM mutants. I would also recommend making the label for this mutant something else as TM is also transmembrane domain. Similarly these experiments should also be preformed in figure 3 and should be shown ass primary figures with the CHO cell data being supplemental. One acknowledges that it is excessively burdensome to do everything in cells where there is no expression of endogenous proteins, however this needs to be done for the major experiments as it is clear that the CHO system is artefactual and should not be the primary cell system used.

The discussion should be adapted so that the major points of the manuscript are easier to understand. If the authors perform the experiments above then the paradigms discussed in this paper would generalizable to both the b1 and b3 integrins and it may be considered a mechanism that works across most integrins.

How does this activation step intersect with talin-mediated FA formation? It would be nice to see an expanded discussion of Talin-mediated FA formation to "speculate" how talin, kindlin, and paxillian might assemble more than 2 integrins. Current discussion seems limited to two integrins for simplicity sake, yet significance of these finds quickly expands if we can see how it amplifies FA assembly. Can authors conclude how these mechanisms may limit FA assembly, i.e. can you speculate which components are essential, i.e. rate limiting?

Is it still correct to distinguish between "micro" clustering and ECM-induced FA formation. It seems data herein is defining the first steps in "macro" clustering that are not necessarily ECM induced. This is found in line 424-28.

More minor comments include:

Line 110: a supplementary figure to support the logic of the chosen talin mutations would be of benefit to the reader.

Line 119: be consistent with significant digits in reporting data. Text has 2 significant figures. Yet figure depicts 3 significant figures, further, 39.5nM rounds to 40 nM, not 39 nM.

Line 498: define buffer composition.

Line 522: HSQC, how were the spectra referenced?

Line 445-48: consider rephrasing, logic of this sentence is difficult to follow.

Reviewer #3 (Remarks to the Author):

This manuscript uses combined biochemical, structural and cell biological techniques to provide a detailed investigation on the role of kindlin protein in the integrin activation by talin-H. The study is remarkably detailed and informative. The authors find that in the integrin activation mechanism the talin-R plays a crucial role in forming a novel talin dimer-paxillin-kindlin supramolecular complex. In particular, they show that talin-R drastically enhances the talin-H-integrin activation by dimerizing the talin in the active form and bridging it with the integrin co-activator partner kindling-2 via the adaptor paxillin protein. Additionally, the data suggest that talin-R contributes to the inside-outside signal pathway

by regulating the binding of the integrin to multivalent ligands.

The main weakness of this work is the NMR-based structural characterization of the interaction between various talin-R fragments and paxillin protein. There are a number of problems that are listed below

1) The authors report in the manuscript a quantitative description of the talin-R fragments and paxillin interactions by SPR measurements indicating that talin-R1-R4 is the principal region involved in the paxillin binding. However, they report that also talin-R9-R12 as well as talin-H display binding to paxillin. Overall, they find 7 subdomains including talin-F3, talin-R2, R3, R4, R8, R11 and R13.

After that, they investigated the paxillin/talin-R fragment by acquiring ¹H-¹⁵N Heteronuclear Single Quantum Coherence (HSQC) spectra focusing the attention on the 1-160 region of the paxillin 1-160 region. First, the material and methods section regarding the experimental NMR procedures (i.e acquisition and processing parameters; molar ratio between the two partners) is completely absent. Additionally, how did the author assign the chemical shifts? Did the authors use previously assigned chemical shifts? If a novel chemical shifts assignment has been performed the authors have to deposit the shifts in the Biomolecular Resonance Bank (BMRB) and report the BMRB ID code in the manuscript. Yet, the chemical shifts analysis reported for each domain is inappropriate (table S1). NMR is a powerful technique to describe per-residue conformational and dynamics features. Therefore, the authors have to i) include in the material and methods a detailed description of the procedure used to analyze the chemical shifts describing the used equation; ii) the plots showing the per-residue chemical shift perturbations for each of the talin-R fragments. It is not clear how the authors classified the data reported in the table S1. In all ¹H-¹⁵N HSQC spectra illustrated in figures the chemical shifts assignment needs to be reported.

2) Moreover, if the talin-R1-R4 is the main portion in the regulation of the paxillin binding (K_D= 171 nM) why didn't the authors use this portion for the NMR titration experiments?

3) Additionally, to provide a more complete description of the Talin/ paxillin, excluding that the isolated talin-R fragments may interact with the same paxillin binding site, the authors have to also see the binding from the paxillin side. In other word they have to perform NMR titration experiments with ¹⁵N-labeled 1-160 paxillin and unlabeled TalinR fragments or better talinTM (active form). These additional investigations will demonstrate their conclusion that the recognition process of talin-R by paxillin occurs via a multi-site binding mode as is already mention in the manuscript without any speculation. Moreover, to fully understand how the paxillin protein acts as link between talin and kindlin structural data regarding this supramolecular machinery is required. Therefore, the authors to address this latter point might perform the same NMR experiments adding also the kindlin protein.

4) Regarding the binding of the ¹⁵N-talin with two paxillin derived peptides the authors have to provide per-residue structural data reporting the chemical shift variations versus residue number. Additionally, they have to provide a more realistic structure and not only a simple structural model generated by sequence alignment and homology modelling (Figure 4D). It is well known that the NMR is one of the most powerful technique for the structure determination. In this case, the author at least for one peptide have to calculate the three-dimensional structure of the complex using CS (Chemical shifts)- or NOE-derived approach.

5) To confirm that talin-R2, R8 and R11 play important roles in the binding to paxillin the authors designed LD binding defective mutants named talin-R2 AKEE; R8 AKEE and R11 TKEE. In the figure 4C the ¹H-¹⁵N HSQC spectra acquired for R2 WT and for the mutant talin-R2 AKEE with and without unlabeled 1-160 paxillin are reported. From the comparison of the pictures (figures 4C and 4E) it seems that the mutations induce significant structural variations (i.e the dispersion of resonances in both dimensions (proton and nitrogen) for the wt and the mutant are quite different). So far, the authors have to provide evidence that the mutations don't alter the 3D fold of the domain by comparing the chemical shift variations between wt/ R2 AKEE. The use of Ca chemical shifts would greatly help this structural analysis.

Minor points.

- 1) In the figures 4C and 4E the axis labels have to be corrected in ^{15}N (ppm) and ^1H**
- 2) In the figure S5 the authors in the figure legend report Free $^{15}\text{N},^2\text{H}$ -talin-R2. Is the protein deuterated? If it is the case the protein is fully or partially deuterated?**

In general, this work has the potential for high impact, as appropriate for publication in Nature Comm. However, the lack of high resolution structural data as well as the incomplete and/or inappropriate analysis of the NMR data, need to be addressed.

General remarks of the revision: We greatly appreciate the reviewers' effort in evaluating our work. We have carefully gone over all the comments and made substantial effort to perform more experiments and revise our manuscript. The revision resulted in revised Fig 1B, revised Fig 3, new Fig 4, new Fig 5D, revised Figs 6F-G (swapped with new Fig S12D), new Fig 7C, new Fig S4, new Fig S5A, new Fig S6C-F, revised Fig S7, new Fig S9, and new Fig S10. Because the change of the text is extensive, we highlighted major changes in the text in red. We hope the revision will be satisfactory since it addressed every question raised by the reviewers.

Reviewer #1 (Remarks to the Author):

1. For the SPR experiments, in the method section it is mentioned that each test was repeated, but I do not see the replicated data. How many times each test/concentration was repeated? The repeated response curves should be provided, and the mean and standard deviation (error range) of kinetic results based on the repeated measurements should be calculated. If the measurement were not repeated, please remove the repeat statement in method, and report the standard deviation of the fitting results (how well is the data fitting to the model).

Response: Thanks for the suggestion. All SPR experiments were indeed repeated at least twice with similar binding kinetics. The error range of each affinity is now provided in all revised SPR figures. We note that many experiments were repeated at different periods with different CM chips and non-identical immobilization conditions, resulting in somewhat different response curves that cannot be exactly aligned but the binding kinetics were similar so we provide one representative sensorgram for each figure with the error range indicated on all SPR figures.

2. Also for the SPR experiments, the paxillinFL was measured at 10 degree C (Fig. S5F), while the rest were measured at room temperature. The binding kinetics is a function of temperature based on thermodynamic principle. Have the authors take this temperature variation into consideration? The weaker KD of 350 nM between Talin-TM/paxillin could be due to this lower temperature measurement. I suggest adding this point to the discussion in page 13.

Response: We did the paxillinFL binding to talin-TM at 10 degree C because we feel these full length proteins may be more stable at lower temperature during the whole assay. However, we just found the sample is fine for the SPR experiments at room temperature as long as we keep the uninjected samples at 10 degree C during the assay and the affinity is indeed stronger at higher temperature as reviewer predicted. To be consistent with other SPR data, we now provide the new data acquired at room temperature (now new Fig S7F).

Reviewer #2 (Remarks to the Author):

1. General:... Experimentally the major issue is the use of an artificial alphaIIb beta3 integrin system in a CHO cell. This system is useful but also has some major problems especially with respect to understanding physiological relevance. These authors have talin double null cells so the manuscript would be significantly improved if they were used for more of the physiological

readouts. In addition, the manuscript would benefit from significant revisions to extrapolate their findings more fully. Further, the details provided explaining the critical first steps of focal adhesion formation are arguably the most significant contribution of this paper and could be showcased more efficiently.

Response: We thank the reviewer for the positive evaluation of our study and helpful comments to further improve it. We also appreciate the reviewer's valuable and critical thoughts on the CHO cell system. We agree that the CHO cell system has some drawbacks to precisely reflect the physiological processes but it is a useful system, as reviewer also agreed, to gain initial insight into the mechanism of integrin activation as demonstrated by many previous studies that were confirmed by subsequent in vivo analyses. Notably, it was this system that led to the discovery of talin as the key integrin activator (Calderwood et al. *J. Biol. Chem*, 274:28071-41999, cited >800 times), which is now widely accepted based on extensive in vivo studies. In our manuscript, the CHO cell system also led us discover the talin dimer-paxillin-kindlin (TPK) machinery in promoting integrin binding to multivalent ligands. We then used structural biology tools to determine the binding mode of the TPK complex and designed key mutation spots to disrupt the complex. These mutations allowed us to verify, again using the CHO cell system, the importance of this complex in regulating integrin activation. Finally, we used talin1/2 null fibroblasts to verify the physiological importance of the TPK complex in integrin activation and post-integrin activation events using the mutations we examined in CHO system. Thus, as you can see, these combined steps were highly integrated to effectively and definitively define the role of TPK in regulating the integrin activation.

Based on the reviewer's suggestions, we made vigorous effort to perform more experiments in talin 1/2 null fibroblasts and then significantly revised the manuscript by putting our findings into a broader context (see also detailed responses to comments 2 and 3 respectively). We also extended our model shown in Figure 7C indicating that the TPK-induced integrin microclusters not only regulate integrin activation but further function as seeds to initiate focal adhesion formation.

Major comments that would significantly improve the study include:

2. In figure 1 what happens to cell adhesion, spreading and integrin activation of beta 1 integrin in the double null fibroblasts with the Tln TM mutants. I would also recommend making the label for this mutant something else as TM is also transmembrane domain.

Response: Thanks for the suggestion. We have now functionally compared talin1/2 null fibroblasts expressing either talin-H (tlnH), talin-WT (tlnWT) or talin-TM (now labeled as tln-M3). Most importantly, while talin-H shows low potency in inducing cell adhesion as expected, expression of tln-WT and tln-M3 induced very similar cell adhesion and spreading, indicating that the retrovirally expressed talin-WT becomes activated by the endogenous activation machinery and does not require activating mutations. This is strikingly different from the CHO model and shows, as indicated by the reviewer, that the fibroblasts represent the more physiological cell system, but both data from CHO cell system and fibroblasts clearly indicate the importance of talin-R since talin-H clearly has low activity compared to either tln-WT or tln-M3. The new data are now shown in Figs 4A, 4B, and S3I. As tln-WT and tln-M3 both facilitate integrin mediated functions in a similar manner in talin1/2-null fibroblasts, the paxillin-binding defective talin mutant

constructs, which were later expressed in talin-1/2null fibroblasts (Figs 4 and 6), were only done using the tln-WT backbone.

As mentioned above, we have changed our nomenclature for talin-TM as tln-M3.

3. Similarly these experiments should also be performed in figure 3 and should be shown as primary figures with the CHO cell data being supplemental. One acknowledges that it is excessively burdensome to do everything in cells where there is no expression of endogenous proteins, however this needs to be done for the major experiments as it is clear that the CHO system is artefactual and should not be the primary cell system used.

Response: Please see our responses to point 1 (General) and point 2. We agree that the CHO cell systems has some drawbacks and therefore, as suggested by the reviewer, conducted a series of new experiments with our talin1/2 null fibroblasts. However, we still would like to keep the CHO cell data within Fig 3 as the PAC-1 binding assays also provide insights into the activity state of the α 5 β 1 integrin, while the fibroblast assays address more general integrin-mediated adhesive activity. Thus, both models complement each other. Our new adhesion data on fibroblasts (new Fig 4) fully support the CHO data shown in Fig. 3. In particular, new Fig 4D shows that the synergistic function of talin and kindlin-2 on cell adhesion depends on the presence of the talin-R domain because talin-WT has much higher capacity to mediate cell adhesion than talin-H whereas in the absence of kindlin-2 talin-WT and talin-H exhibit no difference in mediating cell adhesion. We also specifically investigated the talin-paxillin-kindlin-2 complex by expressing a paxillin-binding deficient kindlin-2 mutant (GLKE) in kindlin-2 deficient fibroblasts, which resulted in reduced adhesion compared to cells that express kindlin-2 WT (new Fig 4F). Moreover, siRNA knockdown of paxillin, while reducing fibroblast adhesion in kindlin-2 WT expressing cells, did not further reduce adhesion when the paxillin-binding mutant kindlin-2 was expressed (new Fig 4F). These data confirmed our initial CHO data showing the importance of the paxillin-kindlin-2 interaction for integrin-multivalent ligand binding and cell adhesion. We also knocked out kindlin-2 in mouse embryonic MEF cells (MEFs) and observed significant adhesion defect in cells expressing the paxillin binding defective kindlin-2 mutant (GLKE) compared to the kindlin-2 WT (Fig S4A). We note that we also attempted to knock-out paxillin in MEFs but the cells displayed abnormal cell morphology and somehow showed dramatically reduced transfection efficiency of paxillin. We then switched to siRNA knock-down method to reduce endogenous paxillin in MEF cells and this approach allowed rather efficient paxillin transfection. Fig S4B shows that expression of paxillin WT but not K2 binding defective paxillin F577E mutant induced the ligand binding/cell adhesion, which cross-confirmed the K2 GLKE knock-in data in K2-KO cells (Fig S4A). Overall, our data clearly showed that disruption of paxillin-kindlin interaction in fibroblasts impairs ligand binding to integrin and cell adhesion. Overall, new Figs 4 and S4 provide strong evidence about the talin-paxillin-kindlin axis in regulating cell adhesion (see lines 219-251 in red for detailed description), which is fully consistent with the CHO cell data.

With regards to the analysis of focal adhesion formation, we would like to point out that our main goal in this study is to understand how integrin activation is regulated by talin-paxillin-kindlin, which has not been reported before. Furthermore, the mutations that disrupt kindlin-2/paxillin interaction have already been shown to impair focal adhesion in our earlier study (ref 56). On the other hand, we showed in Fig 6 that disrupting talin/paxillin interaction impaired focal adhesion formation and cell spreading, which has never been reported so far. Together, all these data are consistent with our CHO cell-based data about the essential role of talin, paxillin, kindlin in integrin activation, cell adhesion and focal adhesion formation.

4. The discussion should be adapted so that the major points of the manuscript are easier to understand. If the authors perform the experiments above then the paradigms discussed in this paper would be generalizable to both the $\beta 1$ and $\beta 3$ integrins and it may be considered a mechanism that works across most integrins.

Response: Almost the entire discussion section has been rewritten (in red) to better and more clearly reflect our points. As suggested by the reviewer, we also proposed that the mechanism on $\beta 1$ and $\beta 3$ integrins derived from this manuscript may be operative on other integrins.

5. How does this activation step intersect with talin-mediated FA formation? It would be nice to see an expanded discussion of Talin-mediated FA formation to “speculate” how talin, kindlin, and paxillin might assemble more than 2 integrins. Current discussion seems limited to two integrins for simplicity sake, yet significance of these finds quickly expands if we can see how it amplifies FA assembly. Can authors conclude how these mechanisms may limit FA assembly, i.e. can you speculate which components are essential, i.e. rate limiting?

Response: See the revised discussion and also revised Fig 7C where we show that integrin microclusters induced by the talin/paxillin/kindlin complex formation not only promotes the integrin binding to multivalent ligands but also serve as seeds for the formation of focal adhesions. We feel that the expression level of paxillin, which is lower than talin and kindlin, may be critical for regulating the ternary complex assembly and determining the growth of FAs (see lines 488-493).

6. Is it still correct to distinguish between “micro” clustering and ECM-induced FA formation. It seems data herein is defining the first steps in “macro” clustering that are not necessarily ECM induced. This is found in line 424-28.

Response: This is an important point. Our description of “micro” clusterings refers to their early occurrence and small scale, in comparison to the “macro” clusterings (FAs) that are normally visible with standard microscopes and require at least one hour to form. To further distinguish the two, we indicated in the discussion (lines 483-487) that microcluster formation is mostly triggered and interlinked via intracellular signaling and adaptor proteins, which prepares cells for efficient cell adhesion (not necessarily binding to ligands). But as an early and critical event, it will also impact following events, for example, cell adhesion and spreading, while these experiments can only be conducted on ECM ligand coated surfaces.

More minor comments include:

7. Line 110: a supplementary figure to support the logic of the chosen talin mutations would be of benefit to the reader.

Response: Revised Fig 1B shows clearly how M319, T1767, and E1770 are located in the autoinhibitory interface and their mutations would disrupt the interface to activate talin.

8. Line 119: be consistent with significant digits in reporting data. Text has 2 significant figures. Yet figure depicts 3 significant figures, further, 39.5nM rounds to 40 nM, not 39 nM.

Response: Those data are all changed to 3 digits now.

9. Line 498: define buffer composition.

Response: Buffer composition added.

10. Line 522: HSQC, how were the spectra referenced?

Response: Please see the revised method section on NMR experiments.

11. Line 445-48: consider rephrasing, logic of this sentence is difficult to follow.

Response: The sentence has been rewritten.

Reviewer #3 (Remarks to the Author):

The main weakness of this work is the NMR-based structural characterization of the interaction between various talin-R fragments and paxillin protein. There are a number of problems that are listed below

1) ... First, the material and methods section regarding the experimental NMR procedures (i.e acquisition and processing parameters; molar ratio between the two partners) is completely absent. Additionally, how did the author assign the chemical shifts? Did the authors use previously assigned chemical shifts? If a novel chemical shifts assignment has been performed the authors have to deposit the shifts in the Biomolecular Resonance Bank (BMRB) and report the BMRB ID code in the manuscript. Yet, the chemical shifts analysis reported for each domain is inappropriate (table S1). NMR is a powerful technique to describe per-residue conformational and dynamics features. Therefore, the authors have to i) include in the material and methods a detailed description of the procedure used to analyze the chemical shifts describing the used equation; ii) the plots showing the per-residue chemical shift perturbations for each of the talin-R fragments. It is not clear how the authors classified the data reported in the table S1. In all 1H-15N HSQC spectra illustrated in figures the chemical shifts assignment needs to be reported.

Response: We apologize that our NMR method section was too brief. This section has been substantially expanded with more experimental details. The molar ratio between the two partners was actually provided in original figure legends of 4C, 4E, S4F, S6, and S7G (now 5C, 5E, S6, S8, S9B-C, S10A-E). We also mention this now in the method section so that readers will know where to find the information.

Regarding chemical shift assignment, we wish to clarify that our major goal of acquiring HSQC of all talin-R subdomains was to verify the binding results derived from pull-down experiments and more specifically define which subdomain is involved in binding to paxillin. HSQC spectra of each subdomain in the absence and presence of paxillin 1-160 as pair were sufficient to reveal the binding based on chemical shift change of the paired spectra so there is no need to know the chemical shift assignment. Using this approach, we found that R1, R5, R6,

R7, R9, R10, R12 had no chemical shift changes (not shown) whereas six other subdomains including R2/R3/R4/R8/R11/R13 had chemical shift changes upon addition of paxillin 1-160. Among these six subdomains, R2/R8/R11 had much more significant chemical shift changes by paxillin (Figs 5C/5E, S5A-B) than R3 and R4 (included as new Fig S6C-D) and R13 had very tiny/almost negligible chemical shift changes by paxillin (new Fig S6E). Based on this observation, we classified R2/R8/R11 as main paxillin binding sites (++) and R3/R4/R13 as weak paxillin binding sites and summarized them in Table S1. The strengths of the R2/R8/R11 binding to paxillin 1-160 were further quantified later in our SPR experiments with the affinities ranging from 2 to 13 μ M (Figs S7A-7C). We revised the text to more clearly describe our approach (lines 281-289).

Since the chemical shifts of R2 and R8 have been published before (BMRB code 17350 and 19339, respectively), we followed the reviewer's suggestion to perform the chemical shift mapping analysis using the equation provided in the method section. New Figs S9B and C show the chemical shift perturbation patterns of R2 and R8 by paxillin 1-160. Since R8 was shown to bind paxillin LD homolog peptide DLC-LD (PDB 5FZT), we hypothesized that LD motifs (Fig S9A) in paxillin 1-160 play important roles in binding to the talin-R subdomains. Indeed, their chemical shift perturbation patterns by LD1 or LD2 (contained in paxillin 1-160) were similar to those of paxillin 1-160 (new Figs S9A-B). The binding results of talinR2/R8/R11 and their mutants to LD1 and LD2 are summarized in Table S2. New Figs S9D-E further show the perturbed LD binding surfaces on R2 and R8, which are similar, and the latter is fully consistent with the known crystal structure of talin-R8 in complex with DLC-LD peptide (Fig S9F). Remarkably, key residues in the LD binding surface of R8, which is structurally similar to R2 and R11, are conserved (Figs S9F vs 5D and S9G), allowing us to design point mutations to disrupt these Paxillin 1-160 LD/talin-R subdomain interaction (Figs S5E-F, S11A-D, and see also Table S2). These data demonstrate that paxillin LD motifs play crucial role in binding to talin-R. We have revised the text to reflect these changes (lines 328-364).

2) Moreover, if the talin-R1-R4 is the main portion in the regulation of the paxillin binding ($K_D=171$ nM) why didn't the authors use this portion for the NMR titration experiments?

Response: The molecular weight of talin-R1-R4 is >60 kDa beyond the range for conventional NMR titration studies/chemical shift assignments (typically <30 kDa). However, we were able to perform the titration studies in reverse way as also suggested by the reviewer, i.e., titrating unlabeled talin-R1-R4 into $^{15}\text{N}/^2\text{H}(80\%)$ -labeled paxillin 1-160 (~18 kDa) (See Figs S10A-D and also response to comment 3 below).

3) Additionally, to provide a more complete description of the Talin/paxillin, excluding that the isolated talin-R fragments may interact with the same paxillin binding site, the authors have to also see the binding from the paxillin side. In other word they have to perform NMR titration experiments with ^{15}N -labeled 1-160 paxillin and unlabeled TalinR fragments or better talinTM (active form). These additional investigations will demonstrate their conclusion that the recognition process of talin-R by paxillin occurs via a multi-site binding mode as is already mention in the manuscript without any speculation. Moreover, to fully understand how the paxillin protein acts as link between talin and kindlin structural data regarding this supramolecular machinery is required. Therefore, the authors to address this latter point might perform the same NMR experiments adding also the kidlin protein.

Response: This is excellent suggestion. We prepared $^{15}\text{N}/^2\text{H}(80\%)$ -labeled paxillin 1-160, which showed largely unfolded pattern as expected from the structure prediction using alphafold2 (except that LD1/LD2 motifs adopt helical conformations, which are known). Because full length talin-TM is too large (540 kDa as a dimer) and also has very limited solubility ($<30\ \mu\text{M}$), we used more soluble talin-R1-R4 ($\sim 60\ \text{kDa}$), which is the major paxillin binding fragment as determined by our study. Titration of 1:0.5, 1:1, and 1:2 ratios of talin-R1-R4 led to line-broadening or significant chemical shift changes of a dozen residues in paxillin 1-160 (new Fig S10D), demonstrating specific recognition. Deletion of N-terminal LD1 showed that some perturbed residues in paxillin 1-160 spectrum disappeared indicating the involvement of LD1 in binding to talin-R1-R4 (new Fig S10B). Deletion of C-terminal LD2 showed that the perturbation of LD1 retains but another set of perturbed residues in paxillin 1-160 disappeared (new Fig S10C), indicating the involvement of LD2 in binding to talin-R1-R4. Fig S10D provides some zoomed regions that underwent chemical shift changes of either LD1 (green arrow) or LD2 (black arrow) in paxillin 1-160 upon binding of talin-R1-R4. These experiments strongly demonstrate the multi-site binding mode of paxillin 1-160 where LD1 and LD2 are separately engaged in binding to talin-R1-R4. Although the total structure determination of the complex ($\sim 80\ \text{kDa}$) is beyond the NMR limit, we attempted to use Alphafold2 to predict the possible binding mode. Remarkably, the predicted complex structure showed that LD1 is involved in binding to R3 whereas LD2 is involved in binding to R2 (new Fig S10E), providing insight into how simultaneous binding of LD1/LD2 in paxillin 1-160 to talin-R1-R4 leads to high affinity ($\sim 153\ \text{nM}$, Fig S7E) as compared to lower affinity of only one LD motif in paxillin 1-160 to individual domain R2 ($K_D \sim 2\ \mu\text{M}$, Fig S7A).

It would be ideal to demonstrate structurally how full length paxillin bridges talin-TM (dimer, $\sim 540\ \text{kDa}$) and kindlin-2 but the whole complex would be $>800\ \text{kDa}$ in dynamic equilibrium, which is beyond the limit of NMR. CryoEM or cryoTM could be a method for structurally characterizing such big and dynamic complex, but is also challenging and beyond the scope of this study. However, our mapping studies clearly showed that the N-terminal paxillin 1-160 is mainly involved in binding to talin-TM whereas our previous studies showed that the paxillin C-terminal LIM4 is mainly involved in binding to kindlin-2 F0 (ref 56). To further demonstrate such core complex formation, we examined ^{15}N -labeled kindlin-2 F0 binding to full length paxillin in the absence and presence of talin-R1-R4. New Fig S12C shows that while addition of paxillin to ^{15}N -labeled kindlin-2 F0 caused line-broadening of many residues in kindlin-2 F0 due to its interaction with paxillin ($\sim 70\ \text{kDa}$), addition of talin-R1-R4 ($\sim 60\ \text{kDa}$) caused additional line-broadening of many perturbed residues in kindlin-2 F0 apparently due to the talin-R1-R4-binding (to paxillin) induced size increase since kindlin-2 F0 has no interaction with talinR1-R4 as we verified. These data provide strong biophysical evidence for the ternary complex formation and are fully consistent with the cell-based Co-IP data showing the ternary complex formation and the mutation-induced disruption of such complex (Fig 6C). Overall, these data define a solid molecular basis for understanding how paxillin can use different regions to simultaneously engage with talin and kindlin to form the supramolecular complex.

4) Regarding the binding of the ^{15}N -talin with two paxillin derived peptides the authors have to provide per-residue structural data reporting the chemical shift variations versus residue number. Additionally, they have to provide a more realistic structure and not only a simple structural model generated by sequence alignment and homology modelling (Figure 4D). It is well known that the NMR is one of the most powerful technique for the structure determination. In this case, the author

at least for one peptide have to calculate the three-dimensional structure of the complex using CS (Chemical shifts)- or NOE-derived approach.

Response: Thanks for the reviewer's suggestion. In response to this, we have performed structure calculation of R2/LD2 complex using Haddock with chemical shift mapping constraints and conserved interface residues on LD motif (see detailed docking procedure in the method section). We note that although both LD1 and LD2 are highly homologous (Fig S9A), we selected LD2 for docking studies since it induces more chemical shift changes of R2 than LD1 (new Fig S9B) and also AlphaFold2 selectively placed LD2 to bind R2 and LD1 to R3 in the predicted structure of paxillin 1-160 in complex with talin-R1-R4 (new Fig S10E). New Fig 5D shows the representative structure with the lowest energy in the ensemble (See also new table S3 for statistics), which resembles the crystal structure of R8/LD complex (Fig S9F). More importantly, the interface is fully consistent with our AKEE mutation data in Figs 5E-F. R11/LD complex is expected to be similar to R8/LD (Fig S9F) and R2/LD (Fig 5D) due to similar fold and conserved binding surface residues (Figs S9D-G), which was further confirmed by the mutation TKEE that disrupted the R11/paxillin LD interaction (Figs 11A-D).

5) To confirm that talin-R2, R8 and R11 play important roles in the binding to paxillin the authors designed LD binding defective mutants named talin-R2 AKEE; R8 AKEE and R11 TKEE. In the figure 4C the 1H-15N HSQC spectra acquired for R2 WT and for the mutant talin-R2 AKEE with and without unlabeled 1-160 paxillin are reported. From the comparison of the pictures (figures 4C and 4E) it seems that the mutations induce significant structural variations (i.e the dispersion of resonances in both dimensions (proton and nitrogen) for the wt and the mutant are quite different). So far, the authors have to provide evidence that the mutations don't alter the 3D fold of the domain by comparing the chemical shift variations between wt/ R2 AKEE. The use of Ca chemical shifts would greatly help this structural analysis.

Response: Overlay of R2 WT and R2 AKEE show very similar spectra with no sign of unfolding/structural change (see the attached figure R1 below) except that a number of residues

near the mutation sites underwent chemical shift changes as expected for the mutation-induced local chemical environmental changes that are not related to 3D structural change. Gel filtration also showed that the mutant eluted at the exactly the same position as the WT protein. To further help elucidate this point, we refer the reviewer to see the mutation sites (Fig 5D) that are totally surface-exposed, which explains why no gross structural change occurred for the mutant vs WT. Overall, all our mutations were selected based on their surface features yet located in the critical interface between the binding partners (Figs 5D, S9D-G).

Figure R1: Overlay of ¹⁵N-labeled talin-R2 WT and its mutant AKEE showing very similar chemical shift dispersion with small chemical shift changes of some residues on and around the mutation sites, indicating that the structural integrity of the mutant (colored in red) is unaltered compared to that of the WT (colored in black).

Minor points.

1) In the figures 4C and 4E the axis labels have to be corrected in ^{15}N (ppm) and ^1HN

Response: Corrected. Note that previous 4C and 4E are now 5C and 5E.

2) In the figure S5 the authors in the figure legend report Free $^{15}\text{N},^2\text{H}$ -talin-R2. Is the protein deuterated? If it is the case the protein is fully or partially deuterated?

Response: It was described in Fig S6 (now Fig S8) legend where we used 80% deuterated ^{15}N -labeled talin-R9 to do the HSQC studies of talin-R9 binding to talin-F2F3 with and without paxillin 1-160.

In general, this work has the potential for high impact, as appropriate for publication in Nature Comm. However, the lack of high resolution structural data as well as the incomplete and/or inappropriate analysis of the NMR data, need to be addressed.

Response: Thanks for the insightful comment and suggestion. The new structural/NMR data and expanded experimental details indeed substantially improved our manuscript.

REVIEWERS' COMMENTS

Reviewer #2 (Remarks to the Author):

The authors answered all my queries I am happy with the manuscript.

Reviewer #3 (Remarks to the Author):

The authors in the revised version of the entitled manuscript "Mechanism of integrin activation by talin and its cooperation with kindling" by Qin and co-workers addressed all major and minor points highlighted in my comments after the first revision. Overall, the quality of the revised manuscript is significantly improved and it is suitable for publication on Nature Communication.

All reviewers have no more concerns about our revised manuscript. We greatly appreciate their effort and time in evaluating our work.